# Ultrastructural heterogeneity of layer 4 excitatory synaptic boutons in the adult human temporal lobe neocortex

**Rachida Yakoubi[1], Astrid Rollenhagen[1], Marec von Lehe[2,3], Dorothea Miller[2], Bernd Walkenfort[4], Mike Hasenberg[4], Kurt Sätzler[5], Joachim HR Lübke[1,6,7]\***

[1]Institute of Neuroscience and Medicine INM-10, Research Centre Jülich GmbH, Jülich, Germany; [2]Department of Neurosurgery, Knappschaftskrankenhaus Bochum, Bochum, Germany; [3]Department of Neurosurgery, Brandenburg Medical School, Ruppiner Clinics, Neuruppin, Germany; [4]Medical Research Centre, IMCES Electron Microscopy Unit (EMU), University Hospital Essen, Essen, Germany; [5]School of Biomedical Sciences, University of Ulster, Londonderry, United Kingdom; [6]Department of Psychiatry, Psychotherapy and Psychosomatics, Faculty of Medicine, RWTH University Hospital Aachen, Aachen, Germany; [7]JARA Translational Brain Medicine, Jülich/Aachen, Germany

**Abstract** Synapses are fundamental building blocks controlling and modulating the 'behavior' of brain networks. How their structural composition, most notably their quantitative morphology underlie their computational properties remains rather unclear, particularly in humans. Here, excitatory synaptic boutons (SBs) in layer 4 (L4) of the temporal lobe neocortex (TLN) were quantitatively investigated. Biopsies from epilepsy surgery were used for fine-scale and tomographic electron microscopy (EM) to generate 3D-reconstructions of SBs. Particularly, the size of active zones (AZs) and that of the three functionally defined pools of synaptic vesicles (SVs) were quantified. SBs were comparatively small (~2.50 $\mu m^2$), with a single AZ (~0.13 $\mu m^2$); preferentially established on spines. SBs had a total pool of ~1800 SVs with strikingly large readily releasable (~20), recycling (~80) and resting pools (~850). Thus, human L4 SBs may act as 'amplifiers' of signals from the sensory periphery, integrate, synchronize and modulate intra- and extracortical synaptic activity.

**\*For correspondence:**
j.luebke@fz-juelich.de

**Competing interests:** The authors declare that no competing interests exist.

## Introduction

The neocortex of various animal species including non-human primates (NHPs) and humans is characterized by its six-layered structure, the organization into vertical oriented functional slabs so-called cortical columns and a system of long-range horizontal axonal collaterals that connect neurons in the same (intra-laminar) and different (trans-laminar) cortical layers of a given brain area, but also with trans-regional projections to different brain regions (reviewed by *Rockland and DeFelipe, 2018*).

The TLN representing ~17% of the total volume of the neocortex (*Kiernan, 2012*) is regarded as a highly specialized associative brain region roughly subdivided into a superior, medial and inferior gyrus that is highly interconnected with the limbic and various sensory systems. In addition, it is well documented that the TLN represents a homotypic granular, six-layered associative neocortex with cytoarchitectonic similarities to primary sensory cortices but different to heterotypic agranular motor cortex (*von Economo and Koskinas, 1925*; *Vogt, 2009*; *Zilles et al., 2015*; *Zilles and Palomero-Gallagher, 2017*). Hence, the TLN is regarded as a higher-order, but not primary, or early sensory neocortex. The growing interest in the TLN is motivated by its importance in superior brain functions

as audition, vision, memory, language processing, and various multimodal associations with other brain regions (reviewed by *Insausti, 2013*). Moreover, the TLN is also involved in several neurological diseases, most importantly, as the area of origin and onset of temporal lobe epilepsy (TLE; reviewed by *Allone et al., 2017*; *Tai et al., 2018*). TLE is the most common form of refractory epilepsy characterized by recurrent, unprovoked focal seizures that may, with progressing disease, also spread to other areas of the brain. Taken together, the TLN represents an important region in the normal and pathologically altered brain in humans. However, relatively little is known about its neural (but see for example: *DeFelipe, 2011*; *Mohan et al., 2015*; *Varga et al., 2015*; *Molnár et al., 2016*), structural and functional synaptic organization (reviewed by *Mansvelder et al., 2019*).

Synapses are highly specialized entities involved in neural communication at any given network of the brain. However, for the most common type of synapses in the CNS, neocortical synapses, rather little data are available, in particular about their quantitative geometry. More recently, neocortical synapses have been quantitatively described in rodents and NHPs (for example: *Anderson and Martin, 2002*; *Freese and Amaral, 2006*; *Popov and Stewart, 2009*; *Rollenhagen et al., 2015*; *Rollenhagen et al., 2018*; *Bopp et al., 2017*; *Hsu et al., 2017*; *Rodriguez-Moreno et al., 2018*), but less is known about these structures in humans, particularly at the presynaptic site (but see *Cragg, 1976*; *Gibson, 1983*; *Kirkpatrick et al., 2006*; *Alonso-Nanclares et al., 2008*; *Blazquez-Llorca et al., 2013*; *Kay et al., 2013*; *Bernhardt et al., 2013*; *Liu and Schumann, 2014*; *Domínguez-Álvaro et al., 2019*). Such coherent and comprehensive quantitative studies depend on the availability of suitable human brain tissue with highly preserved ultrastructure which can be only guaranteed by access tissue from epilepsy surgery biopsy (*Yakoubi et al., 2019*) but not post-mortem material.

The final goal in investigating SBs and their target structures in humans is to describe the synaptic organization of the cortical column, layer by layer, as exemplified by the TLN. First, L5 SBs were quantitatively analyzed (*Yakoubi et al., 2019*), because this layer represents the major output layer from the neocortex to the sensory periphery. The next logical step was to investigate SBs in L4 of the TLN. In granular primary sensory cortices, L4 is regarded as the main recipient layer for signals from the sensory periphery and thus represents the first station of intracortical information processing (reviewed by *Sherman, 2012*; *Clascá et al., 2016*; but see *Constantinople and Bruno, 2013*). Since the TLN is regarded as a higher-order, but not primary, or early sensory neocortex, here L4 represents a convergent input layer for both thalamo-cortical and cortico-cortical inputs (reviewed by *Insausti, 2013*).

We took advantage of non-epileptic neocortical access tissue provided from TLE surgery, and used high-resolution fine-scale transmission and tomographic EM and subsequent computer-assisted quantitative 3D-volume reconstructions of synaptic structures to look for structural correlates relevant for synaptic transmission and plasticity.

The comparison of L4 (this study) and L5 (*Yakoubi et al., 2019*) SBs showed marked and highly significant layer-specific differences in almost all structural parameters investigated. Strikingly, although ~2-fold smaller in SB and AZ size, L4 SBs contain a nearly 3- to 4-fold larger readily releasable (RRP), a 2-fold larger recycling (RP) and a slightly larger resting pool when compared to L5 SBs suggesting marked differences in synaptic efficacy and strength, but also in the modulation of synaptic plasticity. The structural values obtained by *Yakoubi et al. (2019)* and the present study are in line and support recent findings of layer-specific differences in synaptic transmission between L4 and L5 in humans (*Seeman et al., 2018*) and experimental animals (rodent L4 connections: *Feldmeyer et al., 1999*; *Feldmeyer et al., 2002*, reviewed by *Lübke and Feldmeyer, 2007*; rodent L5 connections: *Markram et al., 1997a*; *Markram et al., 1997b*; reviewed by *Ramaswamy and Markram, 2015*). Thus, it might be speculated that layer-specific differences are also present in the other cortical layers to be investigated in humans (work in progress). Hence layer-specific differences in synaptic 'behavior' and computational properties of intra-laminarintralaminar cortical networks could be expected, and may contribute to an 'orchestrated' action of synapses in the entire network of the cortical column.

Finally, the quantitative 3D-models of SBs also provide the basis for realistic numerical and/or Monte Carlo simulations of different parameters of synaptic function, for example, neurotransmitter release and diffusion at AZs that remain only partially accessible to experiment, at least in humans.

## Results

### Density of synaptic contacts established by SBs in L4 of the human TLN

First, the density of synaptic contacts was measured since such data are rare for humans particularly for the TLN (but see *Marco and DeFelipe, 1997*; *Alonso-Nanclares et al., 2008*; *Finnema et al., 2016*). These measurements provide the basis to further gain information regarding the synaptic organization of the neuropil, rate of connectivity as well as possible inter-individual and gender-specific differences in humans (*Table 1 - Source data 1*).

The overall average was $2.37*10^6$ synapses/mm$^3$. The density of synaptic contacts averaged $3.80*10^6$ synapses/mm$^3$ in women (n = 3; ranging from 25 to 36 years of age) and $0.93*10^6$ synapses/mm$^3$ in men (n = 3; ranging from 33 to 63 years in age). Strikingly, a huge inter-individual variability was found: a 1.5- to 4-fold and 2- to 3-fold difference between women and men, respectively.

In all patients, the majority of synaptic contacts counted were excitatory and were found predominantly on spines of different types (76.62%) and on shafts (20.34%). The remainder contacts were inhibitory mainly established on dendritic shafts (1.54%). However, GABAergic terminals were also located on spines, but infrequently (1.33%).

In summary, the synaptic density in L4 of the temporo-basal lobe was ~4-fold higher in women than in men, suggesting a gender-specific difference (see discussion). But the target structure innervation pattern was relatively similar among patients regardless of their gender.

### Neural organization of the human TLN

In granular cortices in rodents and higher mammals including NHPs and humans, L4 is composed of two different types of spiny neurons, the majority of which represent spiny stellate cells with a smaller fraction of star pyramidal cells (*Ahmed et al., 1994*; *Lübke et al., 2000*; *Staiger et al., 2004*; *Egger et al., 2008*; *Oishi et al., 2016*). In Golgi-stained and semithin sections throughout the human TLN, L4 was distinguishable from the lower part of L2/3 and L5 by the absence of pyramidal-shaped neurons (*Figure 1A,B*, *Figure 1—figure supplement 1*) and is further characterized by a zone relatively sparse of neurons with comparatively small, round to ovoid cell bodies organized in a cluster-like fashion (*Figure 1B framed area, D*). At the EM level, GAP-junctional coupling between dendrites was frequently observed (*Figure 1F,G*) in L4, as well as tight junctional coupling mainly between astrocytes. Interestingly, L2 was composed of pyramidal cells that were densely packed but randomly distributed throughout the neuropil in semithin sections (*Figure 1A,B*) whereas L3 contained somewhat larger pyramidal neurons which in contrast to L2 were more loosely distributed (*Figure 1A–C*). Layers 5a, b and 6a were composed mainly of pyramidal neurons of different shape and size (*Figure 1A*, *Figure 1—figure supplement 1*), whereas in sublayer 6b numerous inverted or

**Table 1.** Density of synaptic contacts in L4 of the TLN.

| *Patients | Hu_1 ♀ | Hu_2 ♀ | Hu_3 ♀ | Average ±SD ♀ | Hu_4 ♂ | Hu_5 ♂ | Hu_6 ♂ | Average ±SD ♂ | Total average ±SD |
|---|---|---|---|---|---|---|---|---|---|
| Total density of synaptic contacts/mm$^3$ | $4.1*10^6$ | $1.3*10^6$ | $6.0*10^6$ | $3.80*10^6$ ± $2.36*10^6$ | $0.8*10^6$ | $0.5*10^6$ | $1.5*10^6$ | $0.93*10^6$ ± $0.51*10^6$ | $2.37*10^6$ ± $2.19*10^6$ |
| Excitatory SBs on spines (%) | 60 | 60 | 73.08 | 64.36 ± 7.55 | 100 | 100 | 66.66 | 88.89 ± 19.25 | 76.62 ± 18.75 |
| Excitatory SBs on shafts (%) | 33.33 | 40 | 15.38 | 29.57 ± 12.73 | 0 | 0 | 33.33 | 11.11 ± 19.24 | 20.34 ± 17.75 |
| Inhibitory SBs on spines (%) | 6.66 | 0 | 0 | 2.22 ± 3.84 | 0 | 0 | 0 | 0 ± 0 | 1.33 ± 2.98 |
| Inhibitory SBs on shafts (%) | 0 | 0 | 7.69 | 2.56 ± 4.44 | 0 | 0 | 0 | 0 ± 0 | 1.54 ± 3.44 |

*Patients identity is anonymous due to the new European protection of data privacy (EU) 2016/679.

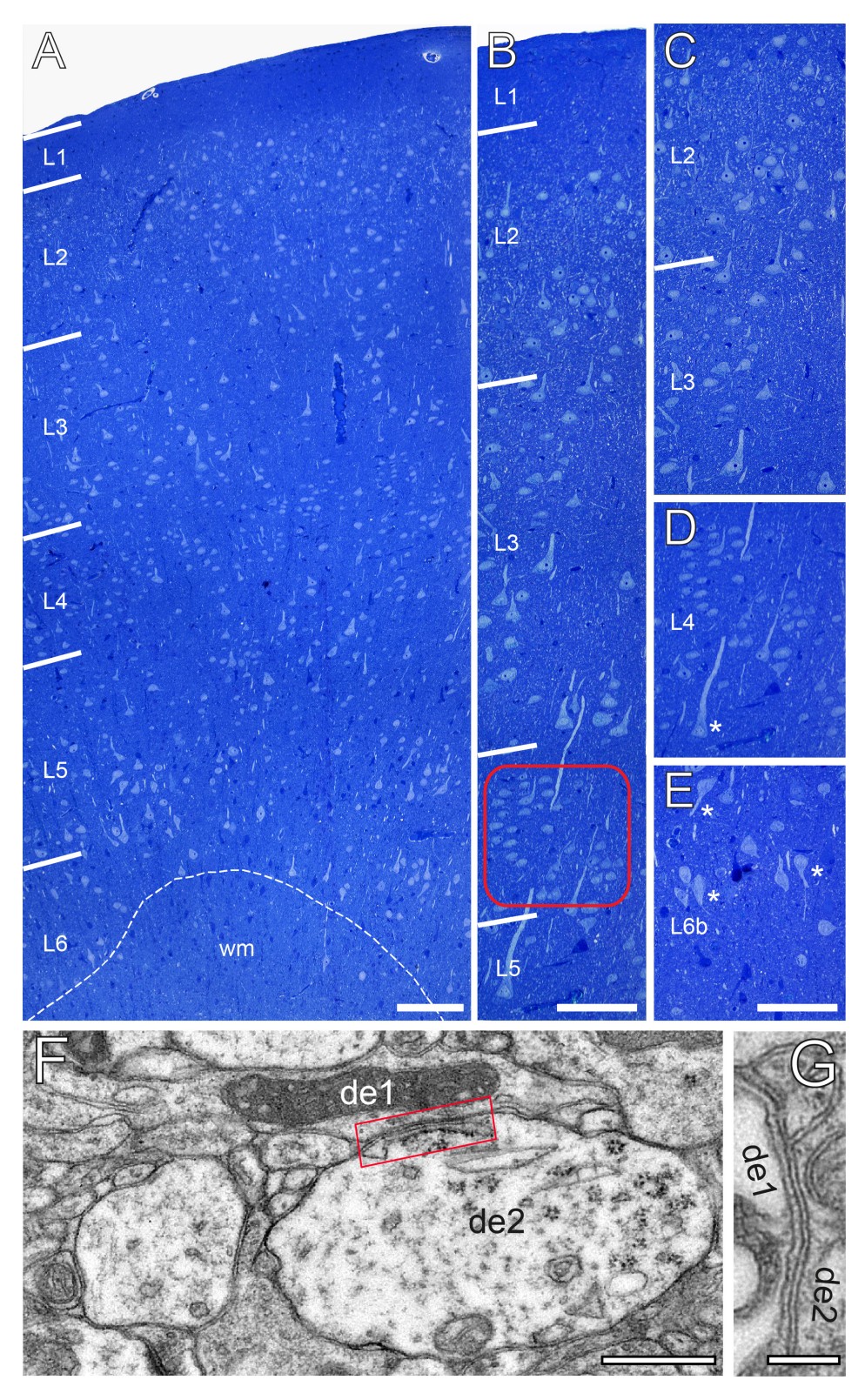

**Figure 1.** The cytoarchitecture of the human TLN. (A) Low-power light micrograph of a methylene blue stained semithin section through the gyrus temporalis inferior (baso-lateral region) in humans, showing the typical six-layered organization of a granular neocortex. Dashed line indicates the gray/white matter border. (B) High-power light micrograph through the same section shown in (A). Note the different density of neurons in L2 and L3. Red frame: region of interest in L4. (C) High-power light micrograph through layers 2 and 3. (D) Cluster-like arrangement of neurons in L4. Note the large L5

*Figure 1 continued on next page*

*Figure 1 continued*

pyramidal cell marked by asterisk. (E) Neuronal organization of L6b containing several inverted pyramidal cells marked by asterisks. Scale bars (A–E) 100 μm. (F, G) Two examples of GAP-junctions between dendrites (de1, de2) one in (F, red frame) and the second in (G) at higher magnification. Scale bars 0.5 μm in F 0.2 μm in G.

The online version of this article includes the following figure supplement(s) for figure 1:

**Figure supplement 1.** Golgi-Cox impregnation of the TLN.

horizontally oriented pyramidal neurons are found (*Figure 1E*) as also described in rodents (*Tömböl, 1984*; *Marx and Feldmeyer, 2013*).

## Quantitative analysis of excitatory L4 synaptic complexes in the human TLN

The main goal of this study was to quantify several morphological parameters representing structural correlates of synaptic transmission and plasticity in L4 excitatory SBs in the human TLN. For this purpose, 150 SBs and 155 AZs were completely reconstructed out of five series of 70–100 ultrathin sections/series using biopsy material from TLE surgery (see Materials and methods).

EM investigation of L4 in the human TLN revealed a dense neuropil containing neuronal cell bodies, astrocytes and their fine processes, dendrites and SBs of different shape and size (*Figures 2* and *3*) and traversing apical dendrites of L5 pyramidal neurons (*Figure 1B*) which are much thicker in diameter and were thus not sampled.

Synaptic complexes in L4 were formed by either presynaptic *en passant* or endterminal boutons (*Figure 2*), with their prospective postsynaptic target structures, either a cell body of a neuron, or dendritic shafts of different calibers (17.7%) or spines of distinct sizes and types (thin 58.20%; filopodial 5.70%; mushroom 7.80%; stubby 3.50% and 7.10% were not classifiable). In our samples, SBs were predominantly (~82% of the total) found on dendritic spines,~3% of which are SBs establishing either two or three synaptic contacts on the same spine and numerous contacts with either the same or different dendrites (*Figure 2B,C*). Interestingly, only ~40% of the spines in our sample contained a spine apparatus (*Figures 2B* and *3A,B*), a specialized form of the endoplasmic reticulum.

L4 SBs were small on average, with a mean surface area of $2.50 \pm 1.78$ μm$^2$, and a mean volume of $0.16 \pm 0.16$ μm$^3$, thus ~3-fold smaller in size than those in L5 of the TLN ($p \leq 0.001$; *Table 2*). SBs were oval to round with a form factor, ranging from 0.27 to 0.77 and a mean of $0.56 \pm 0.09$. Beside relatively large SBs (11.54 μm$^2$; 1.09 μm$^3$) also very small ones (0.42 μm$^2$; 0.01 μm$^3$) were sampled and quantified. Hence, a huge variability was observed with respect to the shape and size of SBs as indicated by the large SD, CV and variance (*Table 2*) regardless of their target structures. Interestingly, a high correlation between the surface area and volume of SBs was observed as indicated by the coefficient of correlation ($R^2$; *Figure 5A - Source data 1*).

In larger SBs, several mitochondria (range 1 to 4) of different shape and size ($0.02 \pm 0.03$ μm$^3$) were present, occupying ~13% of the total bouton volume (*Figures 2A, 3B* and *4A*; *Figure 4 - Source data 2*), similar to that percentage found for mitochondria in L5 of the human TLN (~12%, *Table 2*; *Yakoubi et al., 2019*), whereas smaller SBs contained no (*Figures 2A* and *4A,B*; *Figure 4 - Source data 2*) or only a single mitochondrion (*Figures 2A,C* and *3C*). Thus, a strong correlation was found between the volume of the SBs and the volume of mitochondria (*Figure 5B - Source data 2*), suggesting an important role of these structures in the function of the SB (see Discussion). Worth mentioning is the observation of a few degenerating SBs, characterized by their content of distorted organelles, as well as the presence of several apoptotic neurons identifiable by their dark appearance (*Figure 1*), severe distortions of their cytoplasm and the presence of microglia and macrophages, indicative of cell death of these neurons in our biopsy samples (not shown), although their frequency of appearance varied between tissue samples.

## Structural composition of AZs in L4 excitatory SBs in the human TLN

The number, size and shape of the AZs are key structural determinants in synaptic transmission and plasticity. The majority (~97%) of SBs in L4 had only a single (*Figures 2A* and *3*) at most three AZs. Beside very large AZs spanning the entire pre- and postsynaptic apposition zone (*Figure 2A*), also quite small AZs covering only a fraction of the apposition zone were found (*Figures 2* and *3*). The

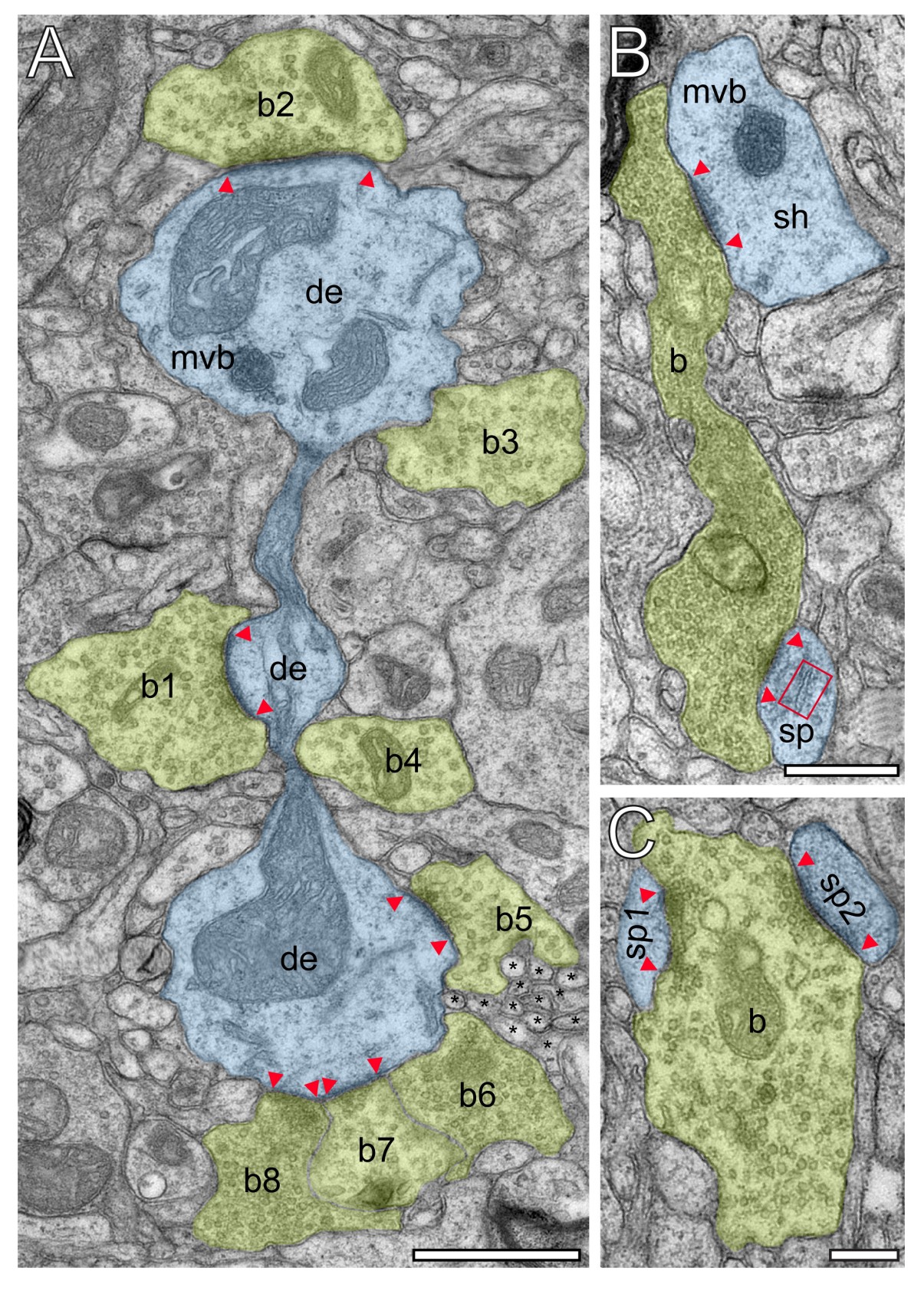

**Figure 2.** Innervation patterns of excitatory L4 SBs and target specificity. (**A**) Dense innervation of *endterminal* SBs (b1-b8; transparent yellow) terminating on a dendrite (de; transparent blue) of ~5 µm length. Note the presence of a cluster of unmyelinated axons (asterisks) isolating SB b5 from SB b6. Scale bar 1 µm. (**B**) *En passant* SB (**b**) innervating a dendritic shaft (sh) and a spine (sp) identified by a spine apparatus (framed area). (**C**) Large *endterminal* SB (**b**) innervating two small spines (sp1, sp2). Scale bars in (**B**) and (**C**) 0.5 µm. Note the presence of multivesicular bodies (mvb) in (**A**) and (**B**). In all images, the AZs are marked by red arrowheads. In (**B**) and (**C**) same color code as in (**A**).

majority of AZs (~63%) showed either a perforation in the PreAZ and PSD or both (*Figures 3B* and *4D - Source data 2*); the remainder AZs were non-perforated (*Figures 2* and *3A,C*). Interestingly, 38.46% of the perforated AZs were established on spines featuring a spine apparatus.

On average, PreAZs were 0.13 ± 0.07 µm² and PSDs 0.13 ± 0.07 µm² in surface area, ranging from 0.03 to 0.39 µm² (PreAZs), and from 0.03 to 0.41 µm² (PSDs) with numerous AZs that were ~2-fold larger than the mean. Interestingly, the size of PreAZs and PSDs in L5 of the human TLN were nearly 1.5- to 2-fold larger (*Table 2 - Source data 2*; see also *Yakoubi et al., 2019*). However, as also described for other CNS synapses of similar size, a huge variability in both the shape and size of the PreAZs and PSDs was observed as indicated by the SD, CV and variance (*Table 2 - Source data 2*). Strikingly, only a weak correlation between the mean surface area of SBs and that of PreAZs (*Table 2 - Source data 2*) was observed. Interestingly, the PreAZ and PSD areas forming the AZ overlapped perfectly in size as indicated by the ratio and the correlation factors (1.01 ± 0.14; *Figure 4D - Source data 2*). Finally, only a weak correlation existed between the PreAZs surface area and the total pool of SVs (*Figure 5F - Source data 2*) suggesting that both the PreAZ size and the total pool of SVs may be independently regulated from the size of the SBs.

Strikingly, the mean surface area of the PreAZs on spines, regardless of the spine type, was almost identical and not significantly different (p=0.06) when compared to those found on dendritic shafts (0.13 ± 0.07 vs. 0.11 ± 0.05 µm²).

The width of the synaptic cleft was 14.11 ± 0.69 nm for the lateral, and 16.46 ± 1.85 nm for the central region and thus significantly narrower when compared with cleft width measured at L5 AZs of the human TLN (*Table 2 - Source data 2*). No clear difference, for example the typical wide broadening of the synaptic cleft, was observed at AZs in our sample, although there is a high variability as indicated by the variance for both the lateral and central edges, respectively (0.04 vs. 0.11).

## Organization of the pools of SVs in L4 excitatory SBs of the human TLN

SVs are the key structures in storing and releasing neurotransmitters, and hence play a fundamental role in synaptic transmission and in modulating short- and long-term synaptic plasticity. Their distribution in the terminal and organization into three distinct functional defined pools, namely the RRP, the RP and the resting pool, regulate synaptic efficacy, strength and determine the mode and probability of release ($P_r$; uni- vs. multivesicular; uni- vs. multiquantal) (*Saviane and Silver, 2006*; *Watanabe et al., 2013*; *Schikorski, 2014*; reviewed by *Schneggenburger et al., 2002*; *Rizzoli and Betz, 2005*; *Neher, 2015*; *Chamberland and Tóth, 2016*).

In general, SVs were distributed throughout the entire terminal in ~85% of the population of SBs investigated (*Figures 2*, *3A,C* and *Figure 4 - Source data 2*). The remainder population of SBs was characterized by rather densely packed or 'clustered' SVs near the PreAZs (*Figure 3B*).

Different types of SVs were found: (1) Small clear SVs with a mean diameter of 19.79 ± 5.62 nm, (2) Large clear SVs with a mean diameter of 69.74 ± 12.26 (3) Mitochondrial-derived vesicles (MDVs) were rarely observed (*Figure 4E - Source data 2*, *Figure 7A*); their role still remains rather unclear. However, they might be involved in the selective transport of mitochondrial substances to lysosomes for degradation (*Sen and Cox, 2017*, reviewed by *Sugiura et al., 2014*; *Misgeld and Schwarz, 2017*) and (4) Large dense-core vesicles (DCVs) with an average diameter of 42.98 ± 14.45 nm. DCVs were seen to either fuse with the presynaptic membrane or intermingled with the population of the SVs throughout the SB (*Figure 4C - Source data 2*). Considering these locations one might already deduce the possible functions of the DCVs, namely: their implication in endo- and exocytosis (*Watanabe et al., 2013*) and build-up of AZs by releasing Piccolo and Bassoon (*Schoch and Gundelfinger, 2006*), or by clustering SVs at the PreAZs (*Mukherjee et al., 2010*; *Watanabe et al., 2013*). In addition various co-transmitters, such as neuropeptides, ATP, noradrenalin, and dynorphin were identified in large DCVs (*Ghijsen and Leenders, 2005*; *Zhang et al., 2011*).

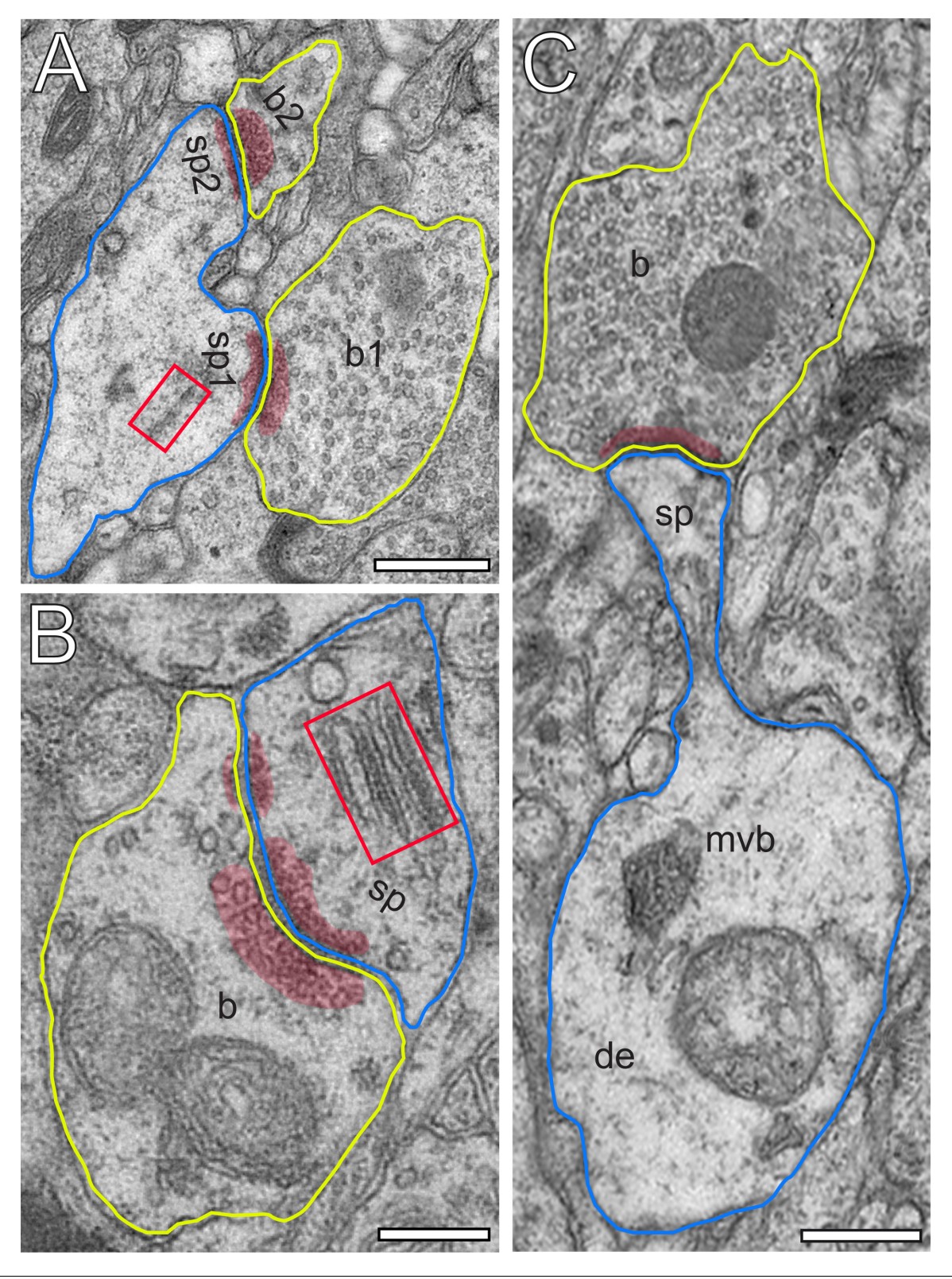

**Figure 3.** Excitatory L4 SBs innervate different types of spines. (**A**) Two SBs (b1, b2; yellow contour) terminating on two stubby spines (sp1, sp2; blue contour). Scale bar 0.5 µm. (**B**) SB (b; yellow contour) on the head of a large mushroom spine (sp; blue contour). Note the presence of two mitochondria occupying a large fraction of the total volume of the SB. Here, SVs were found in closer proximity to the PreAZ. Scale bar 0.25 µm. (**C**) SB (b; yellow contour) terminating on the head of an elongated spine (sp; blue contour) emerging from a relatively large dendritic segment (de; blue contour)

*Figure 3 continued on next page*

*Figure 3 continued*

containing a multivesicular body (mvb). Scale bar 0.5 µm. Note the presence of a spine apparatus (framed area) in (**A**) and (**B**). All AZs are highlighted in transparent red.

**Table 2.** Comparative quantitative analysis of various synaptic parameters in L4 and L5 of the TLN.

| Synaptic boutons | Layers | Mean ± SD | Median | IQR | CV | Skewness | Variance |
|---|---|---|---|---|---|---|---|
| Surface area (µm$^2$) | L4 | 2.50 ± 1.78 | 2.05 | 1.67 | 0.72 | 1.97 | 3.24 |
| | L5*** | 6.09 ± 0.92 | 6.05 | 0.87 | 0.15 | 4.49 | 23.04 |
| Volume (µm$^3$) | L4 | 0.16 ± 0.16 | 0.11 | 0.12 | 1.01 | 2.89 | 0.03 |
| | L5*** | 0.63 ± 0.18 | 0.63 | 0.21 | 0.29 | 2.05 | 0.46 |
| **Active zones** | | | | | | | |
| PreAZ surface area (µm$^2$) | L4 | 0.13 ± 0.07 | 0.11 | 0.08 | 0.54 | 1.35 | 0.005 |
| | L5*** | 0.23 ± 0.05 | 0.22 | 0.07 | 0.22 | 1.86 | 0.03 |
| PSD surface area (µm$^2$) | L4 | 0.13 ± 0.07 | 0.11 | 0.08 | 0.53 | 1.44 | 0.005 |
| | L5*** | 0.29 ± 0.15 | 0.23 | 0.16 | 0.45 | 2.77 | 0.06 |
| **Cleft width (nm)** | | | | | | | |
| Lateral | L4 | 14.11 ± 0.69 | 14.43 | 1.19 | 0.05 | 0.74 | 8.86 |
| | L5*** | 17.25 ± 2.39 | 17.51 | 3.74 | 0.13 | 1.14 | 20.73 |
| Central | L4 | 16.47 ± 1.85 | 15.72 | 3.26 | 0.11 | 0.80 | 17.09 |
| | L5*** | 19.05 ± 2.94 | 18.85 | 2.95 | 0.15 | 1.82 | 30.84 |
| **Mitochondria** | | | | | | | |
| Volume (µm$^3$) | L4 | 0.03 ± 0.04 | 0.02 | 0.02 | 1.04 | 3.71# | 0.001 |
| | L5*** | 0.12 ± 0.09 | 0.07 | 0.16 | 0.87 | 8.22 | 50.30 |
| % to the total volume | L4 n.s. | 13.11 ± 6.20 | 12.78 | 9.25 | 0.47 | 0.17 | 38.47 |
| | L5 | 12.04 ± 1.20 | 11.89 | 2.18 | 0.10 | 0.57 | 23.04 |
| **Synaptic vesicles** | | | | | | | |
| Total number | L4*** | 1820.64 ± 980.34 | 1544.5 | 1119.5 | 0.54 | 0.91 | 961066.59 |
| | L5 | 1518.52 ± 303.18 | 1347.21 | 541.98 | 0.19 | 2.39 | 1655452.24 |
| Diameter (nm) | L4 | 19.80 ± 5.63 | 18.00 | 0.28 | 3.41 | 2.10 | 31.69 |
| | L5*** | 36.69 ± 1.71 | 37.02 | 3.26 | 0.04 | −2.07 | 153.21 |
| Volume (µm$^3$) | L4 | 0.01 ± 0.01 | 0.01 | 0.01 | 1.28 | 3.95# | 0.0002 |
| | L5*** | 0.05 ± 0.02 | 0.05 | 0.03 | 0.4 | 3.60# | 0.002 |
| **Pool size of SVs** | | | | | | | |
| Putative RRP at p10 nm | L4*** | 20.20 ± 18.58 | 17 | 27.25 | 0.92 | 1.11 | 345.04 |
| | L5 | 5.42 ± 4.09 | 4.93 | 6.29 | 0.75 | 2.17 | 39.93 |
| Putative RRP at p20 nm | L4*** | 48.59 ± 39.02 | 41 | 53 | 0.80 | 1.17 | 1523.14 |
| | L5 | 15.21 ± 9.02 | 13.55 | 16.34 | 0.59 | 2.06 | 206.69 |
| Putative RP 60–200 nm | L4*** | 382.1 ± 248.23 | 313 | 376.79 | 0.65 | 1.41 | 61617.55 |
| | L5 | 181.86 ± 27.05 | 180.89 | 47.42 | 0.15 | 1.25 | 11469.97 |
| Putative resting pool > 200 nm | L4 | 1251.82 ± 471.17 | 541 | 471.17 | 0.38 | 1.70 | 87678.29 |
| | L5 n.s. | 1264.07 ± 301.77 | 1150.76 | 540.39 | 0.24 | 0.66 | 72853.49 |

Summary of various structural parameter measurements provided from the detailed 3D-volume reconstructions of SBs in L4 (present study) and L5 (***Yakoubi et al., 2019***) of the human TLN. Mean ± SD, Median, Interquartile Range, CVs, Skewness and Variance were given for each parameter in all patients investigated. #: Values with a skew >3 indicating non-normal distributions. Abbreviations: p10 nm, p20 nm: perimeter 10 and 20 nm from AZ (see Materials and methods), n.s: not significant, ***p≤0.001.

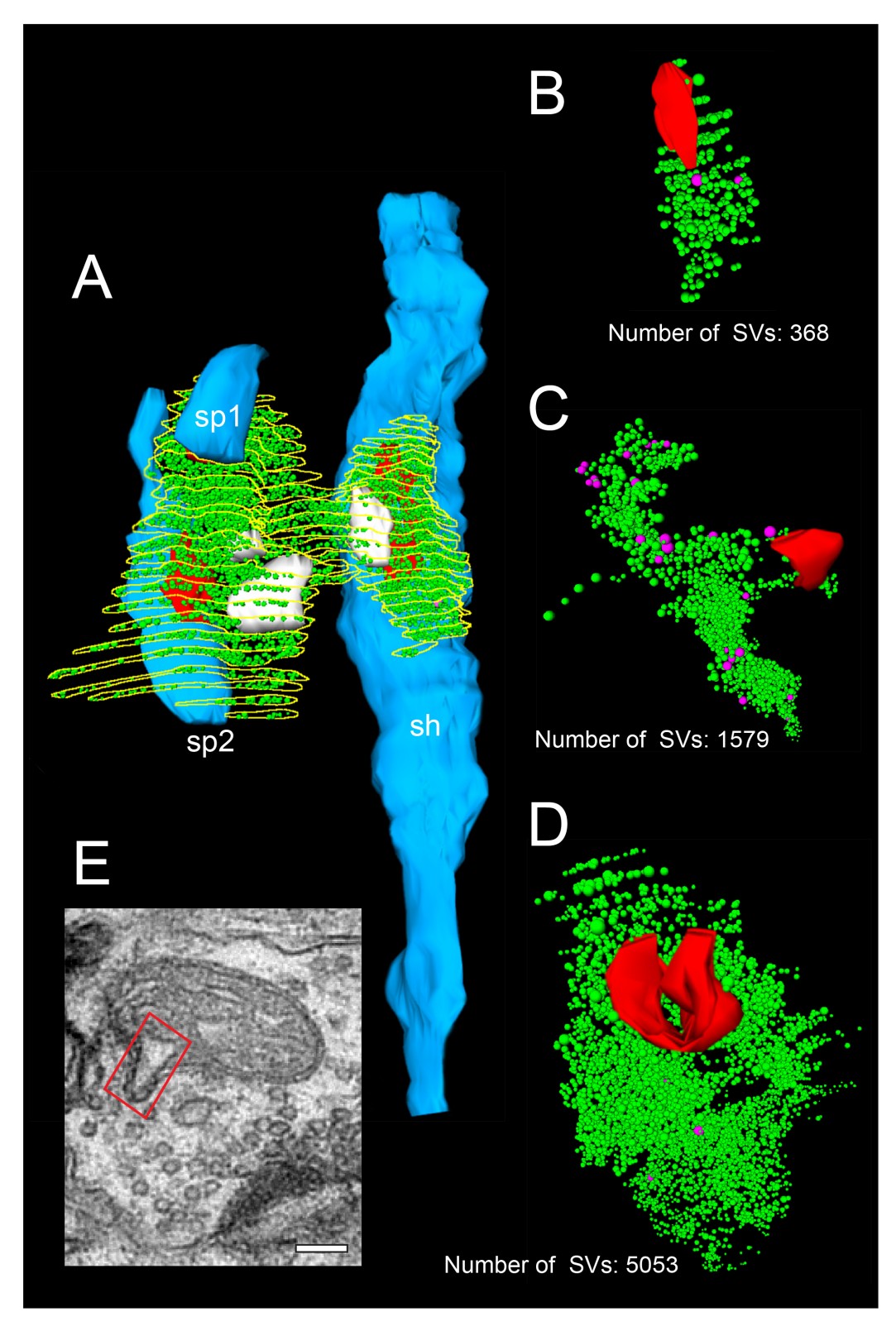

**Figure 4.** 3D-volume reconstructions showing the high variability in the size of the total pool of SVs. (**A**) 3D-volume reconstruction of a SB (yellow outline) innervating a dendritic shaft (sh; blue) and two spines (sp1, sp2; blue). Note the relatively large total pool of SVs (green dots); the size of the PreAZs (red) and the mitochondria (white) always associated to the pool of the SVs. (**B–D**) 3D-volume reconstructions of individual total pools of SVs
*Figure 4 continued on next page*

*Figure 4 continued*

(green dots) at either non-perforated (**B, C**) or perforated (**D**) AZs. Large DCVs (magenta) were frequently observed. (**E**) Example of a mitochondrial-derived vesicle (framed area) in the process of separation from the mitochondrion. Scale bar 0.1 μm.

In our study, the average total pool of SVs was 1820.64 ± 980.34 SVs (ranging from 368 to 5053) occupying ~7% (0.01 μm$^3$) of the total volume of SBs, although the range and the SD indicated a huge variability in total pool size (*Table 2 - Source data 2*) at individual SBs. Strikingly, the total pool in the human TLN was already ~3-fold larger when compared with L4 and L5 SBs in rats (*Rollenhagen et al., 2015*; *Rollenhagen et al., 2018*), and comparable, but significantly different (p≤0.001) from L5 SBs in the human TLN (*Table 2 - Source data 2*; see also *Yakoubi et al., 2019*), although only a low correlation was found between the total pool of SVs and PreAZ surface area (*Figure 5E - Source data 2*) as well as volume of SBs (*Figure 5G - Source data 2*) implying that the total pool of SVs was independent from the size of the SBs.

The distribution pattern of SVs made it impossible to morphologically distinguish the three functionally defined pools of SVs, except for the 'docked' vesicles primed to the PreAZ (reviewed by *Rizzoli and Betz, 2005*; *Denker and Rizzoli, 2010*; *Chamberland and Tóth, 2016*). One method to overcome this problem is to perform a distance (perimeter) analysis that determined the exact location of each SV from the PreAZ. Thus, we assumed that the RRP was located at a distance (perimeter p) of ≤10 nm and ≤20 nm from the PreAZ representing 'docked' and primed SVs fused to the PreAZ. The second pool, the RP, is constituted by SVs within 60–200 nm, which maintained release on moderate (physiological) stimulation. The resting pool, consisted in all SVs further than ≥200 nm, preventing depletion upon strong or repetitive stimulations, but which under normal physiological conditions remains unused.

Using the same perimeter criteria as at human L5 SBs (*Yakoubi et al., 2019*; for criteria see also *Rizzoli and Betz, 2005*), the RRP/AZ was extremely large with an average of 20.20 ± 18.57 at p10 nm and increased by nearly 2.5-fold (48.59 ± 39.02) at p20 nm and thus ~4-fold (p10 nm) and ~3-fold (p20 nm) larger when compared to L5 SBs in the human TLN (p≤0.001, *Table 2 - Source data 2*). However, both pools were characterized by a large variability as indicated by the SD, CV and variance (*Table 2 - Source data 2*) similar to values for L5 SBs suggesting differences in $P_r$, synaptic efficacy, strength and paired-pulse behavior at individual SBs. Interestingly, no correlation was found for the p10 nm and p20 nm RRP with the surface area of PreAZs (*Figure 5H - Source data 2*) in contrast to CA1 synapses (*Matz et al., 2010*).

The RP/AZ was also comparatively large with 382.01 SVs at 60–200 nm but also with a large variability at individual SBs (see the SD and CV) and ~2-fold larger than that in L5 SBs in the human TLN showing a much smaller variability (*Table 2 - Source data 2*). The resting pool contained on average 1438.63 SVs and showed the same variance in pool size as the RRP and RP, but comparable with that estimated in L5 SBs in the human TLN.

However, no correlation was found between the RRP at p10 nm (*Figure 6A - Source data 2*) and p20 nm (*Figure 6B - Source data 2*) and the total pool of SVs minus the RRP at p10 nm and p20 nm, respectively. A weak correlation was observed for the RP (p60-p200 nm) and the total pool of SVs (*Figure 6C-E - Source data 2*). Finally, no correlation existed for the resting (p500 nm) and the total pool of SVs (*Figure 6F - Source data 2*), similar to findings in L5 excitatory SBs in the human TLN (*Yakoubi et al., 2019*).

Taken together, although small in surface area and volume, SBs in L4 of the TLN have strikingly large RRPs, RPs and resting pools when compared with L5 SBs in the human TLN (*Yakoubi et al., 2019*) and CNS synapses of comparable size or even much larger terminals (see Discussion). However, all pools were characterized by a huge variability, were not correlated and hence independent from the SB and PreAZ size.

## EM tomography of L4 excitatory SBs in the human TLN

High-resolution EM tomography was carried out on a sample of small to large SBs (n = 25) with different AZ sizes to look for the organization of SVs, in particular those of the RRP and to test the hypothesis that larger PreAZs display more primed or fused SVs than smaller ones.

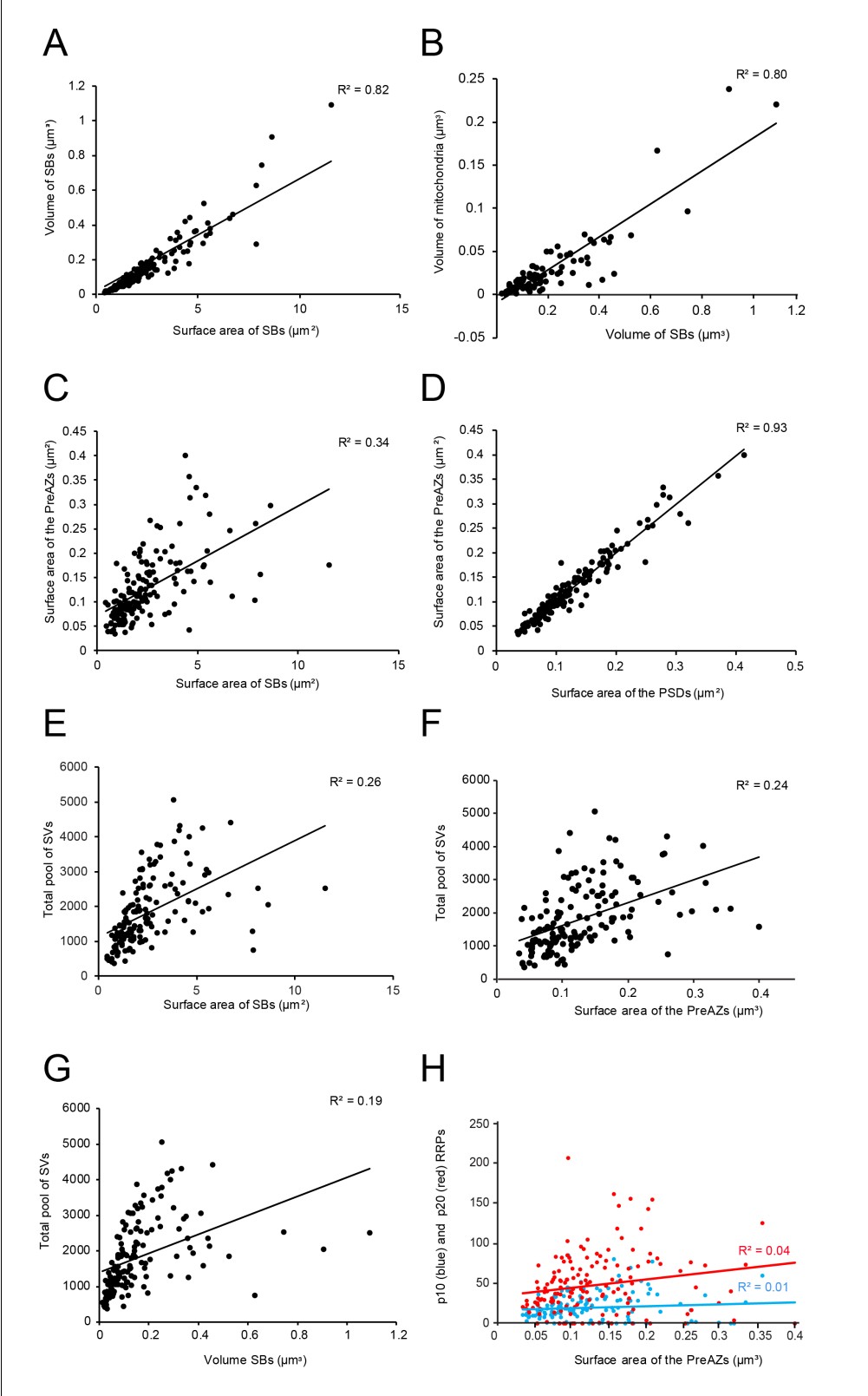

**Figure 5.** Correlations between various structural parameters of L4 SBs. (**A**) The surface area vs. the volume of the SBs. (**B**) The volume of SBs vs. the volume of mitochondria. (**C**) The surface area of SBs vs. the surface area of the PreAZs. (**D**) The surface area of PSDs vs. the surface area of the PreAZs. (**E**) The surface area of SBs vs. the total pool of SVs. (**F**) The surface area of the PreAZs vs. the total pool of SVs. (**G**) The volume of SBs vs. the total pool

*Figure 5 continued on next page*

*Figure 5 continued*

of SVs. (**H**) The surface area of the PreAZs vs. p10 nm (blue dots) and p20 nm (red dots) RRPs, respectively. *Data points were fitted by linear regression and the R$^2$ is given for each correlation.

The results of the tomography were three-fold: First, all SBs analyzed, regardless of their target structures, a dendritic shaft (*Figure 7A*; *Video 1*) or spine (*Figure 7B*; *Video 1*), contained more than one, the most seven 'docked' SVs and/or omega-shaped bodies, already fused (*Figure 7C–F*) with the PreAZ membrane and opened to release neurotransmitter. On average 5.53 ± 0.84 SVs were found at individual PreAZs which was nearly 4-fold smaller when compared to the results of our quantitative perimeter analysis for the p10 nm criterion (20.20 ± 18.58 SVs; see also *Table 2 - Source data 2*). Secondly, there was a tendency that larger PreAZs contained more 'docked' vesicles (*Figure 7A,B*); providing a larger 'docking' area allowing the recruitment of more SVs. However, in a few cases also SBs with a smaller PreAZ were found that had the same number of 'docked' vesicles (*Figure 7C,E*). Finally, so-called MDVs and clathrin-coated pits were clearly identified at several SBs (*Figure 4E - Source data 2*, *Figure 7A*, *Video 1*) thus supporting and substantiating the findings with transmission EM (TEM).

## Cluster analysis of excitatory SBs in L4 of the human TLN

The cluster analysis (CA) revealed two groups of SBs according to their structural parameters analyzed. The principal component analysis (PCA) showed two principal components (PCs) (PC1, PC2) explaining the most variance (*Figure 8A - Source data 2*; *Source code 1–3*); where AZs and SV pools were the main features (parameters) that predominantly contributed to the PCs. The subsequent hierarchical cluster analysis (HCA) based on the AZs and SV pools revealed two stable groups of SBs as shown in the generated dendrograms (*Figure 8B,C - Source data 2*; *Source code 1–3*), where the dissimilarity between the two groups is indicated by the Euclidean height. In *Figure 8B - Source data 2*; *Source code 1–3*, the SBs in the first group (red cluster) had large AZs with 0.14 ± 0.08 μm$^2$ for both PreAZ and PSD surface area, whereas the SBs in the larger group (blue cluster) had smaller AZ (0.12 ± 0.06 μm$^2$) surface areas.

The clustering according to SV pools led also to two groups of SBs, namely SBs (blue cluster) with 25.38 ± 21.09 SVs at p10 nm; 427.29 ± 288.94 SVs in the RP and 995.88 ± 696.49 SVs in the resting pool, respectively. SBs belonging to the red cluster with a smaller pool size compared to the first group: 18.69 ± 17.58 SVs at p10 nm; 368.85 ± 234.72 SVs at RP and 814.45 ± 484.73 SVs at the resting pool, respectively. Although two different clusters existed that further helped to identify subclasses of SBs according to the structural parameters, excitatory L4 SBs in the TLN were relatively similar (*Figure 8C - Source data 2*; *source code 1–3*).

Thus, the CA revealed that the AZs and pools of SVs were the structural parameters that best characterized the SBs in L4 of the human TLN, clustering them into two major groups accordingly.

## Astrocytic coverage of L4 SBs in the human TLN

Astrocytes, by directly interacting with synapses, play an important role in the induction, maintenance and termination of synaptic transmission and plasticity (*Krencik et al., 2017*; reviewed by *Dallérac et al., 2018*). In L4 of the human TLN, astrocytes and their fine processes formed a dense network within the neuropil of the TLN intermingled with neurons and synaptic complexes, composed of the SBs and dendritic shafts or spines. The majority of individual or multiple synaptic complexes (~80%) were tightly enwrapped by fine astrocytic processes physically isolating them from the surrounding neuropil and from neighboring synaptic complexes as shown in L5 of the human TLN (*Yakoubi et al., 2019*).

Astrocytic fingers reached as far as the synaptic cleft under the pre- and postsynaptic density (*Figure 9A*). Hence, it is most likely that fine astrocytic processes at human L4 synaptic complexes are involved in the uptake of neurotransmitter molecules in the synaptic cleft, regulating their temporal and spatial concentration and limiting intercellular crosstalk mediated by volume transmission.

Strikingly, fine astrocytic processes received direct synaptic input from SBs (*Figure 9B*), as also described for L5 SBs of the human TLN (*Yakoubi et al., 2019*), the hippocampus and cerebellar climbing fibers (reviewed by *Allen, 2014*; *Papouin et al., 2017*). On the other hand they provided

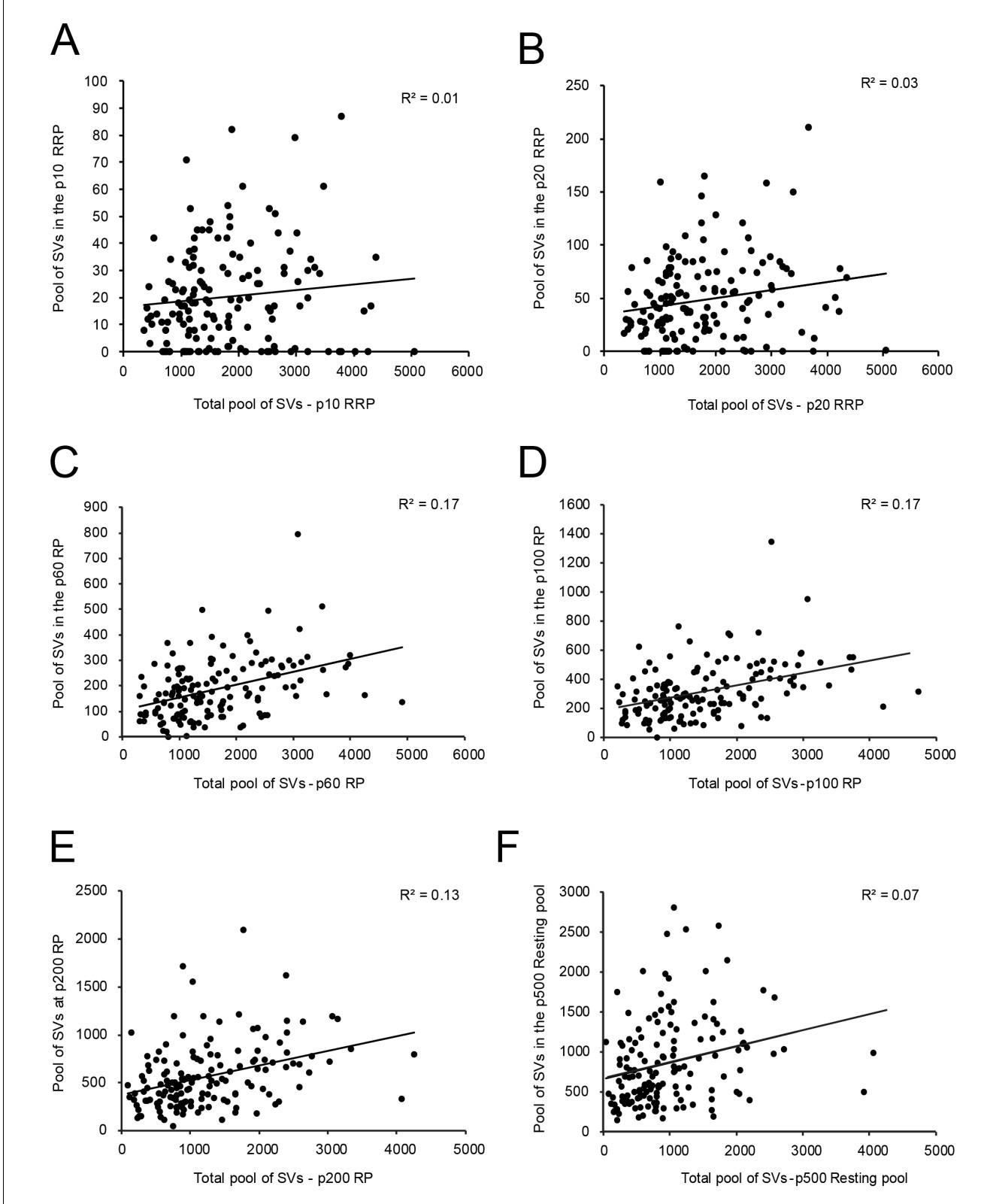

**Figure 6.** Correlations between various structural parameters of L4 SBs and SVs. (**A**) The total pool of SVs vs. the RRP at p10 nm. (**B**) The total pool of SVs vs. the RRP at p20 nm. (**C**) The total pool of SVs vs. the RP at p60 nm. (**D**) The total pool of SVs vs. the RP at p100 nm. (**E**) The total pool of SVs vs. the RP at p200 nm. (**F**) The total pool of SVs vs. the resting pool at p500 nm.

direct input to postsynaptic structures as exemplified by a dendrite in *Figure 9C*. It should be noted that also microglial cells and macrophages were infrequently observed in our tissue samples. However, they can be clearly distinguished from astrocytes and their fine processes by their ultrastructural appearance and the fact that they were never seen to establish direct synaptic contacts.

## Discussion

The present study is the first comprehensive and coherent structural study of L4 excitatory SBs in the human TLN using high-end, fine-scale TEM and EM tomography. Although SBs in any given region of the brain are composed of nearly the same structural subelements, it is their individual and thus specific composition that makes them unique entities, perfectly adapted to their function in the microcircuit in which they are embedded.

Here, we demonstrate marked species and layer-specific disparities of SBs of the human TLN between L4 and L5, in particular in the shape and size of PreAZs and PSDs, and even more importantly in the size of the three functionally distinct pools of SVs (see *Table 2 - Source data 2*). In particular, the size of the RRP and RP in L4 SBs of the human TLN are by 2- to 4-fold larger when compared with L5 excitatory SBs in the human TLN and other SBs of comparable or even larger size in various animal species including rodents and NHPs.

These findings suggest that L4 SBs are strong, efficient and reliable in synaptic transmission. On the other hand, the large variability in the shape and size of AZs and that of the three pools of SVs at individual L4 SBs also suggest a strong modulation of short-term plasticity in the human neocortex (see for example *Varga et al., 2015*; *Molnár et al., 2016*; *Seeman et al., 2018*; reviewed by *Mansvelder et al., 2019*). Hence L4 SBs could act as 'amplifiers' for signals from the sensory periphery, but may also act as 'filters' for incoming information to the TLN.

### Relevance and implications of the density of synaptic contacts measurements

Synaptic density measurement can be a useful tool to not only describe the synaptic organization of a particular area, nuclei and even layers in different brain regions, but also the degree of connectivity underlying the computational properties of a given brain area or network.

Meanwhile several studies in various animal species and brain regions included such an analysis (for example: cat: *Keller et al., 1992*; monkey: *Beaulieu et al., 1992*; *Anderson and Martin, 2002*; *Freese and Amaral, 2006*; *Peters et al., 2008*; mouse: *Merchán-Pérez et al., 2009*; *Bopp et al., 2017*; reviewed by *DeFelipe, 1997*; rat: *Anton-Sanchez et al., 2014*; monkey and mouse: *Hsu et al., 2017*); but data for synaptic density measurements in humans are comparatively rare (but see *Davies et al., 1987*; *Scheff and Price, 1993*; *Blazquez-Llorca et al., 2013*; *Finnema et al., 2016*), in particular for the TLN (*Marco and DeFelipe, 1997*; *DeFelipe et al., 1999*; *Tang et al., 2001*; *DeFelipe et al., 2002*; *Alonso-Nanclares et al., 2008*). Strikingly, a huge difference in the mean density of synaptic contacts was found between our study ($2.37*10^6 \pm 2.19*10^6$) and existing data published by *Alonso-Nanclares et al. (2008)*: $9.13 \pm 0.63*10^8$ and *Tang et al. (2001)*: $164*10^{12}$. This disparities may be attributed to the age, gender, layers, regions investigated and methods used as thoroughly discussed by *DeFelipe et al. (1999)*.

In line with DeFelipe and co-workers (*Alonso-Nanclares et al., 2008*) a marked gender-specific difference was found, however with a higher density of synaptic contacts in women in our study, in contrast to the above cited publication that reported higher values in men; while in NHPs, no gender differences existed (monkey, *Peters et al., 2008*). However, the underlying reasons for gender-specific differences in density of synaptic contacts are difficult to determine, as they may also be influenced by various other parameters, such as hormonal, environmental and emotional factors (for example see *Hyer et al., 2018*).

Our findings may represent the structural correlate to functional gender-specific differences in the brain, for example that women perform better at languages (*Joseph, 2000*; *Kimura, 2000*; *Alonso-Nanclares et al., 2008*); since the TLN is also involved in the memory and language processing and comprehension. Hence, further density measurements are required to demonstrate similarities or differences between the three temporal gyri in the human TL or in comparison to NHPs (but see *Anderson and Martin, 2002*).

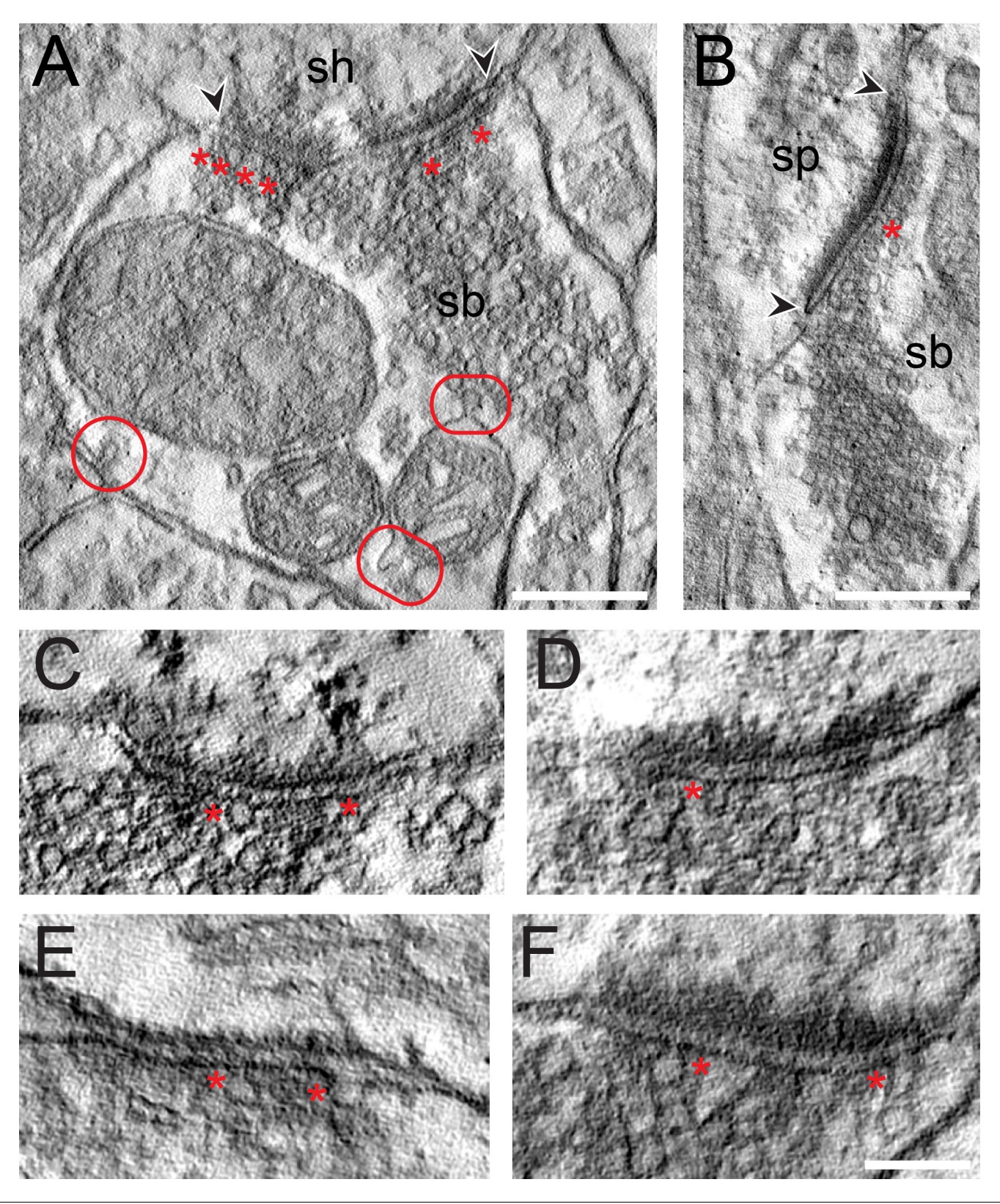

**Figure 7.** EM tomography of L4 SBs in the TLN. (**A**) Example of a SB (sb) terminating on a dendritic shaft (sh) with a large, perforated AZ marked by arrowheads. Asterisks indicate 'docked' SVs. The two frames point to the MDVs and the circle to the clathrin-coated pit. Scale bar 0.25 μm. (**B**) Axo-

*Figure 7 continued on next page*

*Figure 7 continued*

spinous synapse (sb, sp) with a large non-perforated AZ (arrowheads) and an omega-shaped body (asterisk). Scale bar 0.25 μm. (**C, D, E, F**) Four examples of high-power images of AZs where 'docked' SVs or omega-shaped bodies were marked by asterisks. Scale bar 0.1 μm.

In conclusion the TLN exhibited a comparatively high density of synaptic contacts (see also *Finnema et al., 2016*), which together with the large size of the AZs and the three SV pools as demonstrated in the present study imply a high intrinsic and extrinsic connectivity of the TLN, thus shaping the network properties in which the TL is embedded.

## Shape and size of AZs at excitatory L4 SBs in the human TLN

One of the most important structural parameter determining for example synaptic efficacy, strength and $P_r$ is the shape and size of the AZs (*Matz et al., 2010*; *Südhof, 2012*; *Holderith et al., 2012*; *Wilhelm et al., 2014*; *Vaden et al., 2019*). The majority of L4 SBs in the human TLN had only a single AZ as also demonstrated for human L5 SBs and other cortical SBs of similar size in rodents and NHPs (*Marrone et al., 2005*; *Nava et al., 2014*; *Rollenhagen et al., 2015*; *Rollenhagen et al., 2018*; *Bopp et al., 2017*; *Hsu et al., 2017*). Strikingly, the surface area of AZs in L4 SBs was on average 0.13 μm$^2$ and thus 2-fold smaller when compared with L5 SBs in humans (*Yakoubi et al., 2019*), but L4 SBs are among the smallest in the neocortex. However, they were comparable in AZ size with L4 and L5 SBs in rats (*Rollenhagen et al., 2015*; *Rollenhagen et al., 2018*), but ~2- to 3-fold larger than those in mouse and NHP visual, motor and somatosensory neocortex (*Bopp et al., 2017*; *Hsu et al., 2017*). Surprisingly, AZs were somewhat larger than in much bigger CNS terminals such as the Calyx of Held (*Spirou et al., 1998*; *Sätzler et al., 2002*; *Wimmer et al., 2006*), the cerebellar (*Xu-Friedman and Regehr, 2003*); and hippocampal mossy fiber boutons (MFBs; *Rollenhagen et al., 2007*). It has to be noted though that a large variability in both shape and size was observed in all SBs where quantitative data are available.

This variability in AZ size may partially contribute to differences in the mode of release (uni- or multivesicular; uni- or multiquantal release) and quantal size, the size of the RRP and $P_r$ as shown for other CNS synapses (*Matz et al., 2010*; *Freche et al., 2011*; *Holderith et al., 2012*; *Vaden et al., 2019*; reviewed by *Xu-Friedman and Regehr, 2004*). In addition,~63% of AZs showed perforations in their PreAZ and/or PSD, which is higher than reported for L4 (~35%) and comparable with values in L5 SBs (~60%) in rats. A strong correlation between AZs surface area, perforated PSDs and the number of 'docked' and resting pool SVs was reported for rat neocortex (*Nava et al., 2014*; *Rollenhagen et al., 2015*; *Rollenhagen et al., 2018*) and is also supported by the findings in this study. It has to be noted that a large proportion of the AZs in L4 SBs established on spines nearly occupied two-third or even the entire pre- and postsynaptic apposition zone, suggesting that excitatory synaptic transmission is highly efficient at these structures by increasing the docking area for primed and 'docked' SVs which is further supported by our EM tomography experiments (see *Figure 7*). Interestingly, only a weak correlation between the PreAZ surface area with that of the bouton was found in human and rat cortical L4 and L5 excitatory SBs (*Rollenhagen et al., 2015*; *Rollenhagen et al., 2018*; *Yakoubi et al., 2019*; this study), implying that the size of the AZs is an independent structural parameter and may be regulated in an activity-dependent manner as shown for hippocampal SBs in the CA1 subregion (*Matz et al., 2010*; *Holderith et al., 2012*).

Only a weak correlation was found between the surface area of the PreAZ and the total pool of SVs, with no correlation for the p10 nm and p20 nm RRP, respectively. A slight tendency was found that SBs with larger PreAZs contained more 'docked' vesicles which is in contrast to our quantitative analysis concerning the RRP, but in good agreement with our tomography experiments and studies in the hippocampal CA1 region where a direct correlation between the size of the AZ and the number of SVs in the RRP together with an increase in $P_r$ was demonstrated (*Matz et al., 2010*; *Holderith et al., 2012*). However, our sample size with EM tomography was too small to substantiate this correlation for PreAZs in L4 SBs of the human TLN.

Altogether, the relatively large size, perfect overlap and high number of perforated PreAZs and PSDs at L4 SBs in the human TLN may partially contribute to a high $P_r$, and thus reliable synaptic transmission (see *Seeman et al., 2018*). On the other hand, the large variability in AZ size at

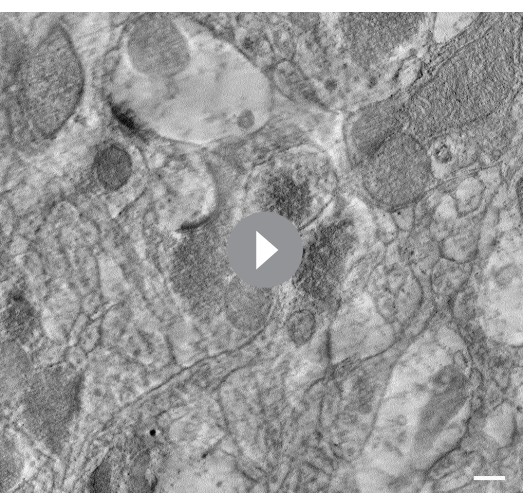

**Video 1.** EM tomography of L4 SBs in the human TLN. Two SBs terminating on dendritic spines, both with a large non-perforated AZ occupying half of the pre- and postsynaptic apposition zone with a relatively large pool of SVs, one of which is clustered around the AZ (lower right corner). Note, also the SB establishing a synaptic contact with a dendritic shaft (upper left corner). All three SBs contain a single but large mitochondrion. Scale bar 0.25 μm.
https://elifesciences.org/articles/48373#video1

individual SBs may play a role in the modulation of synaptic plasticity and paired-pulse behavior at individual SBs.

## Size of the three pools of SVs

During prolonged and intense activity, synaptic transmission could be modulated in various ways depending on the availability of SVs and on their recycling rates. Hence, the size of the RRP critically determines synaptic efficacy, strength and plasticity as described for various CNS synapses (*Rosenmund and Stevens, 1996*; *Schikorski and Stevens, 2001*; *Rizzoli and Betz, 2004*; *Schikorski, 2014*; *Watanabe et al., 2014*; *Rollenhagen et al., 2018*; *Vaden et al., 2019*; reviewed by *Rizzoli and Betz, 2005*; *Neher, 2015*; *Chamberland and Tóth, 2016*).

Although smallest in size around neocortical SBs, L4 SBs in the human TLN had a total pool size of ~1800 SVs/AZ; comparable but significantly larger than in L5 terminals in the human TLN (*Table 2 - Source data 2*), but nearly more than 3-fold (~550 SVs/AZ; *Rollenhagen et al., 2015*), and ~2-fold (~750 SVs/AZ; *Rollenhagen et al., 2018*) larger than their counterparts in L4 and L5 in rats. Comparison with even giant CNS terminals, for example adult MFBs (~20-fold larger in size), revealed a total

pool size of ~850 SVs/AZ (*Rollenhagen et al., 2007*), ~600 SVs/AZ at cerebellar MFBs (*Saviane and Silver, 2006*), and even a nearly 12-fold larger total pool (~125 SVs/AZ) when compared to one of the largest CNS terminals, the Calyx of Held (*Sätzler et al., 2002*).

The already large size of the total pool of SVs in L4 SBs in the human TLN also predicts comparatively large RRPs and RPs. Indeed, the putative RRP was on average 20.20 ± 18.57 (p10 nm) and doubled to 48.59 ± 39.02 (p20 nm) SVs/AZ, ~3- to 4-fold larger than in L5 SBs of the human TLN, ~5-fold larger than those in L5 SBs (p10 nm 3.9 ± 3.4, p20 nm 11.56 ± 4.2; *Rollenhagen et al., 2018*), and 8–10-fold larger than that in L4 SBs (p10 nm 2.0 ± 2.6, p20 nm 6.3 ± 6.4; *Rollenhagen et al., 2015*) in rats, respectively. Comparison with even larger CNS synaptic terminals revealed a more than 12-fold and 8-fold difference (hippocampal MFBs p10 nm 1.6 ± 1.5, p20 nm 6.2 ± 4.1; *Rollenhagen et al., 2007*) and Calyx of Held (p10 nm 1.9 ± 2.0, p20 nm 4.8 ± 3.8; *Sätzler et al., 2002*). Our estimates of the size of the RRP is even more substantiated and supported by our EM tomography. Although the number of 'docked' SVs and omega-shaped bodies was ~4-fold smaller than the average size of the p10 nm RRP, this can be explained by the inclusion of SVs that are not 'docked' but within 10 nm distance from the PreAZ, whereas only 'docked' SVs were counted using EM tomography.

Hence SVs in the RRP are rapidly available to sensory stimulation and execution of complex behaviors as well as Up-states (*Zhou and Fuster, 1996*; *Sanchez-Vives and McCormick, 2000*; *Sakata and Harris, 2009*).

This notion is even more supported by the size of the putative RP/AZ which was ~380 SVs at human L4 SBs, ~2-fold larger as in L5 SBs of the human TLN, ~130 and ~200 SVs in rat L4 and L5 SBs, ~3700 SVs for adult rat MFBs, but nearly 4-fold larger than that reported for the rat Calyx of Held (~60 vesicles). Finally, the resting pool of SVs in L4 SBs is large enough to rapidly replenish the RRP and RP and thus guarantee only a partial depletion even at repetitive high-frequency stimulation.

Taken together, the comparatively large size of AZs and that of the three pools of SVs determines and provide the basis for high reliability in synaptic transmission, efficacy and strength in L4 SBs of the human TLN. The marked differences in AZ and SV pool sizes between individual SBs may

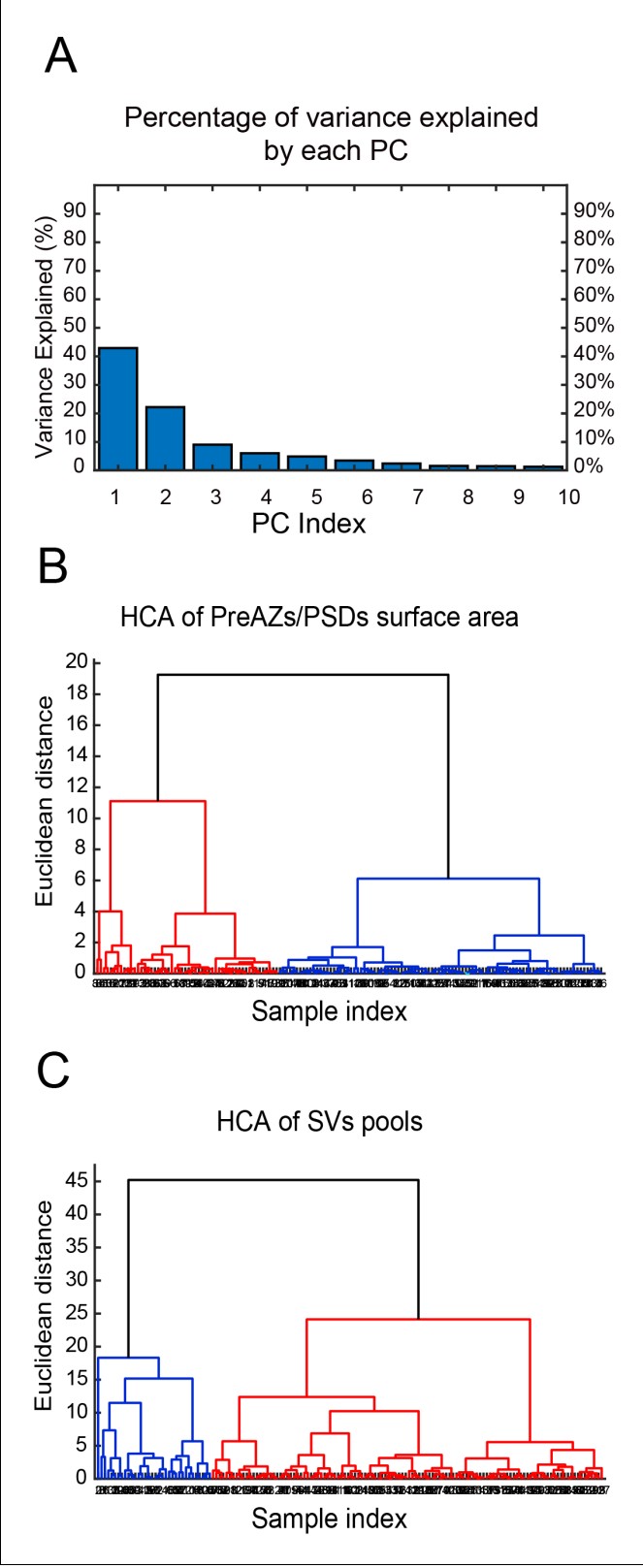

**Figure 8.** CA of PreAZs/PSDs surface area and SVs pools in excitatory L4 SBs in the human TLN. (**A**) Bar histogram showing the PCs of all structural parameters analyzed. (**B, C**) Two dendrograms generated from the CA, identifying two groups (clusters) of L4 SBs according to the PreAZs/PSDs surface area (**B**) and the size of SV pools (**C**). The Euclidian height indicates the difference or dissimilarities between the clusters.

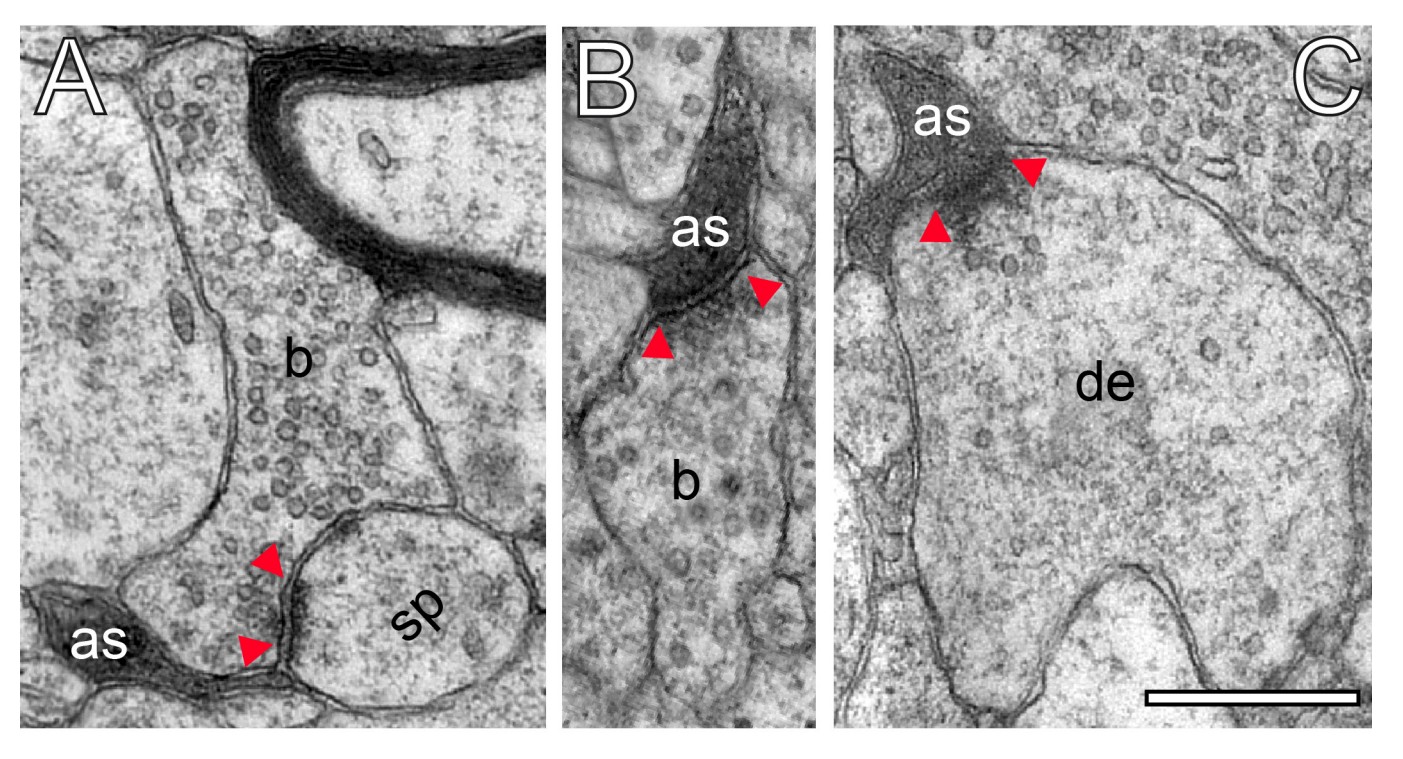

**Figure 9.** Astrocytic interactions. (A) Fine astrocytic process (as) reaching as far as the synaptic cleft at the synaptic apposition zone between a spine (sp) and a SB (b). (B), (C) Direct synaptic contact established between an astrocytic finger (as) with a SB (b) in (B) and a putative dendrite (de) in (C). Note the presence of SVs in the dendrite in (C). Scale bars in (A–C) 0.5 μm. In all images, the AZs are indicated by red arrowheads in (A–C).

underlie rapid changes in the computational properties of single neurons or networks during Up- and Down states in behavior.

## Other important structural components of synaptic complexes in L4 of the human TLN

It has to be noted that in contrast to rat and mouse neocortex, the majority of excitatory L4 SBs (~85%) in the human TLN was established on spines of different types. In addition, not all shaft synapses were GABAergic and in turn not all SBs terminating on spines were glutamatergic, although such contacts were only rarely observed (see also *Kwon et al., 2019*).

## Role of spine apparati in synaptic plasticity

Approximately half of all spines in L4 of the human TLN contained a spine apparatus, a highly specialized derivate of the endoplasmic reticulum, involved in spine motility and the stabilization of the SB and its target structure at the pre-and postsynaptic apposition zone. The high abundance of a spine apparatus confirms and supports findings in rat L4 and L5 (*Rollenhagen et al., 2015*; *Rollenhagen et al., 2018*) and human L5 synaptic complexes in the TLN (~65%; *Yakoubi et al., 2019*), but differs substantially to hippocampal CA1 synapses where only a smaller fraction (~20%) of spines contained these structural subelements (*Martone et al., 1996*; *Spacek and Harris, 1997*; *Deller et al., 2003*). It has been demonstrated that the abundance of a spine apparatus partially contribute in modulating short- and long-term potentiation by stabilizing the axo-spinous complex during the initial high-frequency stimulation (*Holtmaat et al., 2005*; *Umeda et al., 2005*).

## Importance of mitochondria in synaptic transmission

Mitochondria are important structural components present in all CNS nerve terminals but with marked difference in their numbers. At L4 SBs in the human TLN they are often organized in clusters

associated with the pool of SVs, in line with observations at human L5 excitatory SBs in the TLN (*Yakoubi et al., 2019*) and several other CNS synapses in various animal species (for example see *Rowland et al., 2000*; *Wimmer et al., 2006*; *Rollenhagen et al., 2007*; *Rollenhagen et al., 2015*; *Smith et al., 2016*). Mitochondria are reported to be highly mobile (*Mironov, 2006*; *Mironov and Symonchuk, 2006*), act as internal calcium stores (*Pozzan et al., 2000*; *Rizzuto et al., 2000*) and hence regulate internal $Ca^{2+}$ levels in nerve terminals (*Perkins et al., 2010*). More importantly, they are involved in the mobilization of SVs from the resting pool (*Verstreken et al., 2005*; *Perkins et al., 2010*; *Smith et al., 2016*). In excitatory L4 SBs of the human TLN mitochondria contribute to ~13% of the total volume of the boutons suggesting a strong contribution in the induction of several signal cascades, for example in the priming and docking process, relying on the rapid availability of $Ca^{2+}$ in the SB.

## Astrocytic coverage

Finally, the majority of synaptic complexes (~85%) in L4 (this study) and L5 (*Yakoubi et al., 2019*) of the human TLN were tightly enwrapped by fine astrocytic processes reaching as far as the synaptic cleft. This is in line with findings at other small-sized CNS synapses (*Xu-Friedman et al., 2001*; *Rollenhagen et al., 2015*; *Rollenhagen et al., 2018*), but in marked contrast to the hippocampal MFB (*Rollenhagen et al., 2007*), the calyx of Held-principal neuron synapse (*Müller et al., 2009*) and hippocampal CA1 synapses (*Ventura and Harris, 1999*). There, only ~50% were directly found at the synaptic interface (*Ventura and Harris, 1999*) suggesting that hippocampal astrocytes do not uniformly sample glutamate at the synaptic cleft. At hippocampal MFBs and the calyx of Held, fine astrocytic processes were never seen to reach as far as the synaptic cleft. It has been demonstrated that at large CNS synapses glutamate spillover, synaptic cross talk and a switch from asynchronous to synchronous release upon repetitive stimulation occurred at these synapses (*von Gersdorff and Borst, 2002*; *Hallermann et al., 2003*). Astrocytes at L4 synaptic complexes thus act as physical barriers to neurotransmitter diffusion thereby preventing spillover of released glutamate by active take-up and removal of glutamate in the cleft. In addition, they terminate synaptic transmission and may thus, speed-up the recovery from receptor desensitization (*Danbolt, 2001*; *Oliet et al., 2004*). Both mechanisms allow the precise spatial and temporal regulation of the neurotransmitter concentration in the synaptic cleft (*Anderson and Swanson, 2000*).

Furthermore, astrocytes release glutamate or GABA (*Le Meur et al., 2012*) through vesicular exocytosis, which can also regulate synaptic transmission through activation of pre- and postsynaptic receptors (*Haydon and Carmignoto, 2006*). Finally, astrocytes are crucial for the induction and control of spike-time dependent depression (t-LTD) at neocortical synapses by gradually increasing their $Ca^{2+}$ signaling during the induction of t-LTD (*Min and Nevian, 2012*).

Thus, it is most likely that astrocytes at L4 excitatory SBs in human TLN govern synaptic transmission and plasticity by the temporal and spatial modulation of the glutamate concentration profile thereby shaping the EPSP amplitude.

## Functional relevance

In general, L4 in primary sensory cortices is regarded as the main input layer for thalamo-cortical afferents from the respective thalamic relay nuclei (*Ahmed et al., 1994*; *Rodriguez-Moreno et al., 2018*, reviewed by *Clascá et al., 2016*, but see *Constantinople and Bruno, 2013*). Thus, L4 represents the first station of intracortical information processing from peripheral sensory signals. From L4, these signals are then transferred via translaminar connections within a cortical column and via transcolumnar axons to adjacent columns or even transregional to other cortical regions (reviewed by *Rockland and DeFelipe, 2018*). It has to be noted though that the TLN is regarded as a higher-order, but not primary, or early sensory neocortex. Thus, L4 of the TLN represents a convergent input layer for both thalamocortical and corticocortical synaptic inputs.

In rat and mouse neocortex, and recently in humans, it has been elegantly shown by paired recordings that L4-L4 excitatory synaptic connections are characterized by a comparatively high synaptic efficacy and strength as indicated by their high average EPSP amplitudes (rat barrel cortex 1.0 to 1.6 mV, mouse primary visual cortex 0.54 mV, human L4 0.95 mV), low coefficient of variations (<0.4) and low failure rates (<5%) indicative for highly reliable synaptic transmission (rat: *Feldmeyer et al., 1999*; mouse and human: *Seeman et al., 2018*) when compared with excitatory

connections in other cortical layers (reviewed by *Lübke and Feldmeyer, 2007*). Some of these L4-L4 connections in rat somatosensory cortex are so strong that high-frequency trains of postsynaptic action potentials can be evoked causally related to the high abundance of NMDA-receptors contributing with ~40% to the overall EPSP amplitude (*Feldmeyer et al., 1999*; *Rollenhagen et al., 2012*). In addition, L4-L4 excitatory synaptic connections show a high degree of bidirectional coupling (~30%) resulting in recurrent excitation or feed-back inhibition.

Finally, rat L4 spiny neurons are highly interconnected with ~200 other excitatory spiny neurons within a 'barrel' column: in turn ~300–400 L4 spiny neurons converge onto a single L2/3 and L5 pyramidal neuron (*Feldmeyer et al., 2002*; *Lübke et al., 2003*). This intra-columnar connectivity is not only a major determinant for reliable signal transduction in L4, acting as 'feed-back amplifiers' even for weak signals from the sensory periphery, but also for the safe and reliable distribution of signals to other neurons located in different layers within the cortical column with which L4 neurons are interconnected (canonical microcircuit of the cortical column).

Assuming such a scenario described above exists in L4 of the human TLN, several structural parameters contribute to its function as an important associational area involved in the induction, maintenance and regulation of various computations underlying perception, executive control, learning and memory in which the TLN plays an important role. Hence several structural subelements may contribute to reliable signal transduction: The shape and size of AZs and the large number of SVs in the RRP and RP implying a high $P_r$ underlying high synaptic efficacy and strength that contribute to feed-back amplification of even weak sensory signals, and in addition may also enhance TL intracortical information processing. On the other hand L4 may, in addition, act as a 'filter' for incoming signals from the sensory periphery or from other cortical and subcortical areas with which the TLN is interconnected.

The astrocytic coverage preventing glutamate spillover further guarantees a direct control and sharpening of the transmitted signals. On the other hand, the large variability in AZ size and the three pools of SVs may be involved in the sorting, modulation, and further discrimination of intrinsic and extrinsic signals by neurons in the TLN. Together, all these characteristics ensure the proper wiring and firing of neurons in L4 of the human TLN (see *Seeman et al., 2018*), to accomplish its function as a recipient layer for both thalamocortical and corticocortical inputs and help to explain information processing from incoming signals of the sensory periphery, within the TLN and from brain regions with which the TLN is interconnected.

## Perspectives to work on the TLN

As already mentioned earlier, the TLN represents ~17% of the total volume of the neocortex in humans and is regarded as a multimodal associational granular neocortex with a wide range of reciprocal connections with other cortical and subcortical sensory and associational areas. To date it is still rather unknown whether the TLN has a columnar organization and a 'canonical' circuitry of neurons and how this influence its synaptic organization. Hence our first aim was and is to describe the synaptic organization and quantitative geometry of SBs in the TLN layer by layer using high-end fine-scale TEM and EM tomography.

Secondly, the TLN is the area for the onset of TLE, the most common type of epilepsy, which has not been investigated at the subcellular synaptic level, but we will address this question in future studies. Our recent investigations provide the basis to directly compare 'healthy' neocortical access tissue with that from patients that suffer from epilepsy throughout their live and did not underwent epilepsy surgery until recently (work in progress).

Preliminary results show a complete loss in cortical and laminar architecture and thus synaptic organization of an epileptically infiltrated TLN. The superficial part of the neocortex is characterized by a complete lack of neurons, absence of dendritic structures and a marked damage of SBs in the neuropil in two patients. Interestingly, this situation changes gradiently when moving to the more infragranular part of the TLN. Here, more and more SBs become 'normally' structured, neurons and dendritic structures start to reappear in the neocortex, and down to the white matter the neocortex appears 'normally' structured. However, how such a severely damaged neocortex can compensate the complete 'loss' of its supragranular layers that are connected and interact with their counterparts on the opposite hemisphere remains unanswered. How that would influence the synaptic organization and its possible re-arrangement is a challenging task to accomplish.

## Materials and methods

### Human neocortical tissue processing for EM

Biopsy material was obtained from three male and three female patients (25–63 years in age, see *Table 3*) who suffered from drug-resistant TLE und underwent surgery to control the seizures. The consent of the patients was obtained and all experimental procedures were approved by the Ethical Committees of the Rheinische Friedrich-Wilhelms-University/University Hospital Bonn (ethic votum of the Medical Faculty to Prof. Dr. med. Johannes Schramm and Prof. Dr. rer. nat. Joachim Lübke, Nr. 146/11), and the University of Bochum (ethic votum of the Medical Faculty to PD Dr. med. Marec von Lehe and Prof. Dr. rer. nat. Joachim Lübke, Reg. No. 5190-14-15; ethic votum of the Medical Faculty to Dr. med. Dorothea Miller and Prof. Dr. rer. nat. Joachim Lübke, Reg. No. 17–6199-BR), and the EU directive (2015/565/EC and 2015/566/EC) concerning working with human tissue.

During surgery, blocks of neocortical access tissue from the temporo-basal regions of the inferior temporal gyrus (*Figure 10*) were taken far from the epileptic focus and may thus be regarded as non-affected (non-epileptic) tissue as routinely monitored by preoperative electrophysiology and magnetic resonance imaging (MRI). Other evidence that confirms the 'normality' of biopsies and rules out the effect of disease and treatment is the homogeneity of synaptic parameters analyzed among patients as shown by the boxplots (*Figure 11 - Source data 2*). This has also been demonstrated by other recent structural and functional studies using the same experimental approach (*Testa-Silva et al., 2014*; *Mohan et al., 2015*; *Molnár et al., 2016*; *Seeman et al., 2018*; *Yakoubi et al., 2019*).

After their removal, biopsy samples of the TLN were immediately immersion-fixed in ice-cold 4% paraformaldehyde (PFA) and 2.5% glutaraldehyde (GA) in 0.1 Mphosphate buffer (PB, pH 7.4) for 24 to 48 hours (hrs) at 4°C. Vibratome sections (150-200 μm in thickness, VT1000S, Leica Microsystems GmbH, Wetzlar, Germany) were cut in the frontal (coronal) plane through the human temporo-basal neocortex and post-fixed for 30 to 60 min in 0.5 or 1% osmiumtetroxide (Sigma, Munich, Germany) diluted in PB-buffered sucrose (300 mOsm, pH7.4) at room temperature in the dark. After visual inspection and thorough washing in PB they were dehydrated in a series of ethanol starting at 20% (15 min for each step) to absolute ethanol (twice 30 min) followed by a brief incubation in propylene oxide (twice 2 min; Fluka, Neu-Ulm, Germany). Sections were then transferred into a mixture of propylene oxide and Durcupan resin (2:1, 1:1 for 1hr each; Fluka, Neu-Ulm, Germany) and stored overnight in pure resin. The next day, sections were flat-embedded on coated glass slides in fresh Durcupan, coverslipped and polymerized at 60°C for 2 days.

Tissue blocks were examined under the light microscope (LM) to determine the region of interest (ROI). Semithin sections were cut with a Leica UltracutS ultramicrotome (Leica Microsystems, Vienna, Austria), with a Histo-Diamond knife (Fa. Diatome, Nidau, Switzerland) stained with methylene-blue (Sigma-Aldrich Chemie GmbH, Taufkirchen, Germany) to identify the cortical layers, examined and documented with LM using a motorized Olympus BX61 LM microscope equipped with the Olympus CellSense analysis software (Olympus GmbH, Hamburg, Germany). Then, serial (70-100) ultrathin sections (50 ± 5 nm thickness) were cut through L4. ROIs within a series, containing well-preserved structures, were photographed at 8000x with a Zeiss Libra 120 (Fa. Zeiss, Oberkochen, Germany) equipped with a Proscan 2K digital camera (Fa. Tröndle, Moorenweis, Germany) using the

**Table 3.** Summary of patient data

| Patient identity | Gender | Age (years) | Age at epilepsy onset (years) | Histopathological result | Antiepileptic drugs (pre-op) | Reconstructed SBs |
|---|---|---|---|---|---|---|
| Hu_1 | Female | 36 | 4 | GGL | LTG, LEV | 53 |
| Hu_2 | Female | 25 | 12 | AHS | LTG | 25 |
| Hu_3 | Female | 25 | 23 | GGL | Zebinix, LEV | 25 |
| Hu_4 | Male | 33 | 5 | Gliosis | LEV, CBZ | 25 |
| Hu_5 | Male | 63 | 24 | AHS | LEV, LTG, CBZ | 22 |
| Hu_6 | Male | 49 | 36 | AHS | Vimpat, ZNS | - |

AHS: Ammon's horn sclerosis; GGL: Ganglioglioma; LEV: Levetiracetam; LTG: Lamotrigine; ZNS: Zonisamide; CBZ: Carbamazepine.

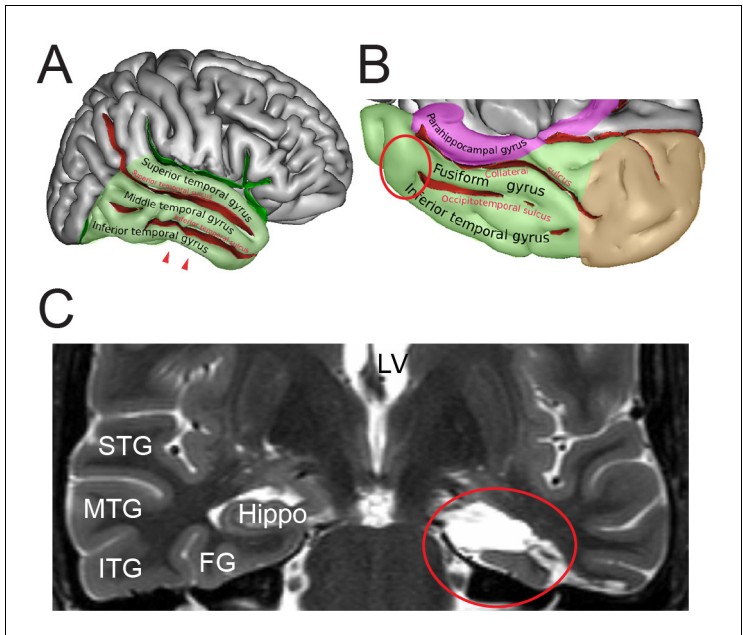

**Figure 10.** Identification of the region of interest in the human TLN. (**A**) Lateral view of the human right cerebral hemisphere. The areas highlighted in transparent green represent the TL. The depth of the sulci are colored in red. The arrowheads indicate the region of interest (ROI) in the inferior temporal gyrus. (**B**) Midsagittal and oblique view of the human right cerebral hemisphere. Color code as in (**A**) Brown represents the occipital lobe and purple the parahippocampal region respectively. The red circle indicates the ROI. Figures (**A**) and (**B**) were retrieved and modified from https://en.wikipedia.org/wiki/Inferior_temporal_gyrus#/media/File:TempCapts.png under the following license: https://en.wikipedia.org/wiki/Creative_Commons. (**C**) Postoperative fMRI after the corresponding epilepsy surgery (selective amygdalohippocampectomy). The sampling site of the biopsy material is circled in red, representing the region between the inferior temporal gyrus and the fusiform gyrus. Abbreviations: FG: fusiform gyrus, Hippo: hippocampus, ITG: inferior temporal gyrus, LV: lateral ventricle, MTG: middle temporal gyrus, STG: superior temporal gyrus.

© Sebastian, 2011. Panels A and B retrieved and modified from https://en.wikipedia.org/wiki/Inferior_temporal_gyrus#/media/File:TempCapts.png under a Creative Commons Attribution-ShareAlike License (CC BY-SA 3.0)

SIS Multi Images Acquisition software (Olympus Soft Imaging System, Münster, Germany). EM images were then edited using Adobe Photoshop and Adobe Illustrator software for publication.

## Golgi-Cox impregnation of biopsy material

Four tissue blocks were processed with the Golgi-Cox impregnation technique using the commercially available Hito Golgi-Cox OptimStain kit (Hitobiotec Corp, Kingsport, TE, USA). After removal of the biopsy samples, tissue were briefly rinsed twice in double distilled water (dd $H_2O$), and then transferred into the impregnation solution overnight at room temperature. The next day, samples were incubated in a fresh impregnation solution and stored for 14 days in the dark at room temperature. Sections were then transferred in solution 3 in the dark at room temperature for one day. Thereafter, sections were placed into a fresh solution 3 in the dark at room temperature for 6 additional days. Then, solution 3 was exchanged and samples were stored at 4°C in the dark overnight. Tissue blocks were embedded in 5% Agarose (Carl Roth, Karlsruhe, Germany) diluted in dd $H_2O$, and sectioned with a vibratome in the coronal plane at 100–250 µm thickness and then transferred to dd $H_2O$. After careful removal of the agarose, free-floating sections were incubated into solution 3 for 2–3 min in the dark at room temperature, and right after placed into dd $H_2O$, washed several times and stored overnight. The next day, sections were put into a mixture of solutions 4 and 5 for 10 min at room temperature. Afterwards, they were rinsed twice in dd $H_2O$ for 4 min each, and dehydrated in 50%, 70% and 95% ethanol for 5 min each, then transferred into absolute ethanol (3 × 5 min), defatted in xylene and finally embedded in Eukitt, (Sigma-Aldrich Chemie GmbH,

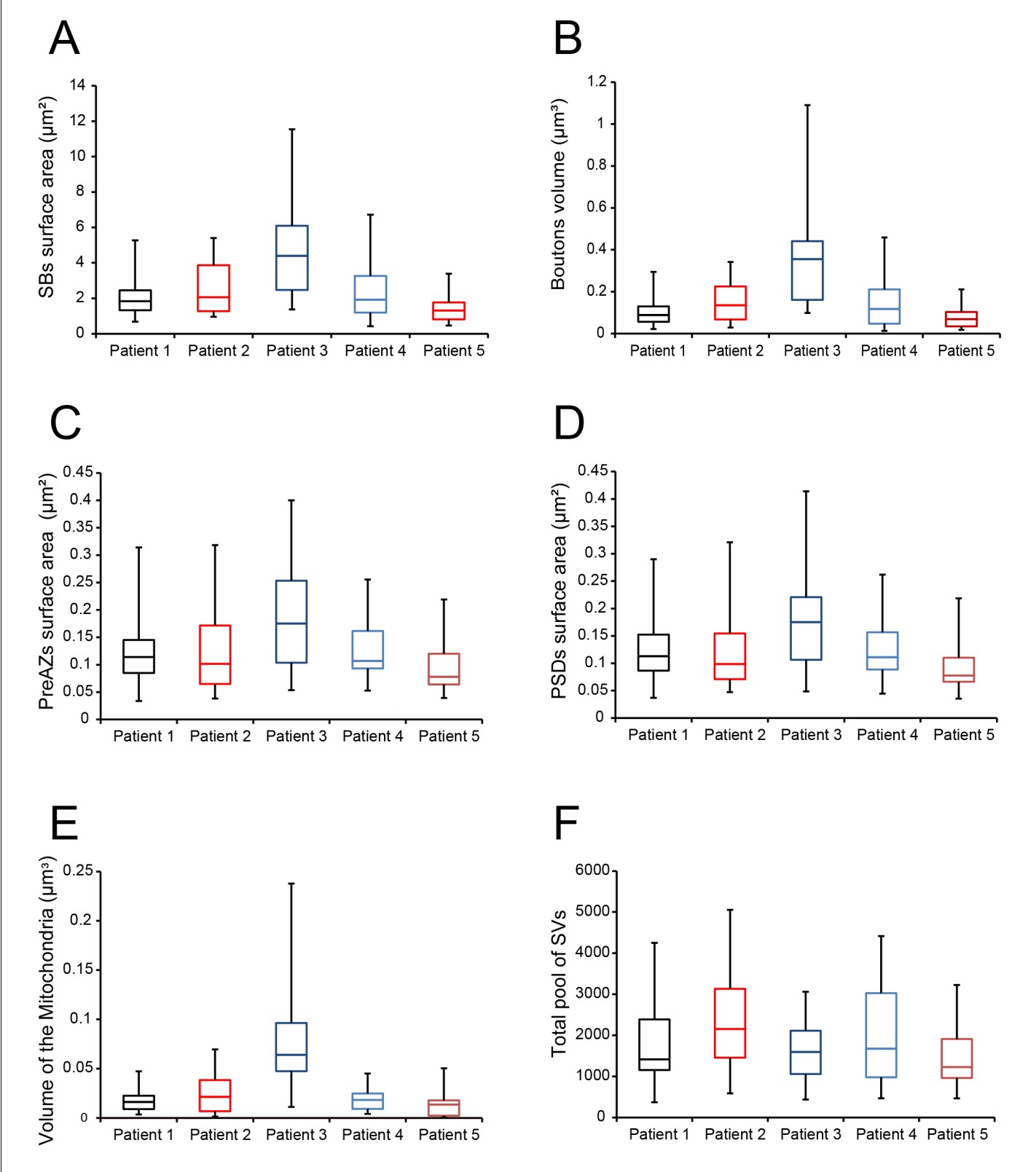

**Figure 11.** Boxplots of various structural parameters. Data distributions for each patient are indicated by the medians (horizontal bars), IQRs (framed areas), minimum and maximum (vertical lines) for the distribution of: (A) Surface area of SBs; (B) Volume of SBs; (C) PreAZ surface area; (D) PSD surface area; (E) Volume of mitochondria; (F) total pool of SVs. Note that most structural parameters are not significantly different.

Taufkirchen, Germany) coverslipped and air-dried. Finally, sections were examined and imaged with an Olympus BX 61 LM equipped with the CellSense software package (Olympus, Hamburg, Germany).

## Stereological estimation of the density of synaptic contacts

The density of synaptic contacts in a given volume is a valuable parameter to assess the structural and functional changes in the brain, which are linked to the age, pathological or experimental conditions (*Rakic et al., 1994*; *DeFelipe et al., 1999*). The density of synaptic contacts was unbiasedly estimated in six patients (see *Table 3*) using the physical dissector technique (*Mayhew, 1996*; *Fiala and Harris, 2001*) by counting the synaptic complexes in a virtual volume generated by two adjacent ultrathin sections that is the dissector: the reference section and the look-up section (*Sterio, 1984*; *Kaplan et al., 2011*). Here, counting was performed using FIJI (*Schindelin et al., 2012*; https://fiji.sc) on a stack of 20 aligned serial electron micrographs for each patient taken from the series of ultrathin sections used for 3D-volume reconstructions of SBs. An unbiased counting frame was first set and synaptic contacts to be considered (counted) are the one present in the reference section only, and meeting the following criteria: presence of a PreAZ and a prominent or thin PSD separated by a synaptic cleft and SVs in the presynaptic terminal. Care was taken to distinguish between excitatory and inhibitory synaptic contacts, as well as the postsynaptic target structures (dendritic spines or shafts). Finally the density of synaptic contacts ($N_v$) per 1 mm$^3$ (see *Table 1* - *Source data 1*) was calculated using the formula below (*Fiala and Harris, 2001*):

$$\mathrm{Nv} = \frac{\sum d\, Qd}{\sum d\, Vd}$$

where $Q_d$ is the number of synaptic contacts per dissector and $V_d$ is the volume of the dissector given by: Number of dissectors x frame area x section thickness.

## 3D-volume reconstructions and quantitative analysis of excitatory L4 SBs

All electron micrographs composing each series were imported, stacked, and aligned in the reconstruction software OpenCAR (Contour Alignment Reconstruction; for details see *Sätzler et al., 2002*). Synaptic structures of interest were outlined on the outer edge of their membranes throughout the series. 3D-volume reconstructions were then generated and the following structural parameters were analyzed: 1) surface area and volume of SBs; 2) volume of mitochondria; 3) surface area of the PreAZs and PSDs; 4) number and diameter of clear synaptic and DCVs, and 5) total pool of SVs and the RRP, RP and resting pool.

Excitatory SBs were characterized by large round SVs and prominent pre- and postsynaptic densities in contrast to putative GABAergic terminals that have smaller, more oval-shaped SVs and thin PSDs. A SB was considered completely captured, when it was possible to follow the axon in both directions through the entire series (*en passant* bouton) or the enlargement of the axon leading to an endterminal bouton. The start of a bouton was defined by the typical widening of the axon and the abrupt occurrence of SVs.

The PreAZs and PSDs were regarded as complete when their perimeters were entirely reconstructed in a series of ultrathin sections. The PreAZ surface area was measured by extraction from that of the presynaptic terminal membrane. The size of the PSD was the perimeter ratio between the outlines of the PSD to that of the synaptic contact. The synaptic cleft diameter was measured because of its importance for the transient increase of glutamate concentration, reversible binding of glutamate to appropriate glutamate receptors and eventual uptake and diffusion of glutamate out of the cleft. To a large extent, these processes are governed by the geometry of the synaptic cleft and the shape and size of the PreAZs and PSDs. Synaptic cleft width measurement was performed only on synaptic contacts cut perpendicular to the AZ. The distance between the outer edge of the pre- and postsynaptic membranes at the center of the synaptic contact and at the two lateral edges was measured and averaged for each synaptic contact. All SVs were marked throughout each SB and their diameters were individually measured. To determine the distribution profile of the SVs, the minimal distance between each SV membrane to the contour lines of the PreAZ was measured

throughout the SB in every single image of the series. Large DCVs were only counted in the image where they appeared largest (for details see *Yakoubi et al., 2019*).

In this study, aldehyde fixation was used that is thought to induce tissue shrinkage thereby biasing structural quantification (but see *Eyre et al., 2007*; *Korogod et al., 2015*). A direct comparison of structural parameters obtained from either aldehyde or cryo-fixed and substituted tissue samples (*Korogod et al., 2015*), had shown differences in cortical thickness (~16% larger in cryro-fixed material), volume of extracellular space (~6-fold larger in cryo-fixed material), a slight increase in glial volume and overall density of synaptic contacts (~14% in cryo-fixed material), but no significant differences in neuronal structures such as axons, dendrites and vesicle length.

Concerning synaptic parameters as estimated here no significant difference was found for SB size and other synaptic subelements such as mitochondria, AZs and SVs (*Zhao et al., 2012a*; *Zhao et al., 2012b*). Therefore, no corrections for shrinkage were applied and we are thus convinced that the synaptic parameters reported here are accurate and can be directly used for detailed computational models. In addition, large-scale preservation for ultrastructural analysis will therefore continue to rely on chemical fixation approaches, due to the limited preservation of the ultrastructure in cryo-fixed material as stated in *Korogod et al. (2015)*.

## Focused ion beam scanning electron microscopy

In this study FIB-SEM was used on L4 of the human TLN to investigate the dynamic changes of the neuropil through a large z-dimension (*Video 2*).

Immediately after explantation, neocortical access tissue samples (n = 2) of the Gyrus temporalis inferior were immersion-fixed in an ice-cold mixture of phosphate-buffered 4% PFA and 2.5% GA for 4 hr. Subsequently, the samples were post-fixed overnight in 0.15M cacodylate buffer (CB) + 2% PFA, 2.5% GA and 2 mM $CaCl_2$ before they were embedded in 4% Agar-Agar dissolved in water. After removing access Agar-Agar, vibratome sections of 150 µm thickness were cut (VT1000S, Leica Microsystems GmbH, Wetzlar, Germany) in the frontal (coronal) plane through the human TLN. Sections were collected in multi-well plates in 0.3M CB + 4 mM $CaCl_2$ and thoroughly washed (5 × 3 min) with 0.15M CB + 2 mM $CaCl_2$. Thereafter, sections were incubated in 0.15M CB + 1.5% potassium hexocyanoferrate (II), 2% osmium tetroxide and 2 mM $CaCl_2$ for 1 hr on ice, in the dark. After washing (5 × 3 min) with deionized water ('MilliQ', Merck Millipore, Burlington, MA, USA), sections were placed in an aqueous 1% thiocarbohydrazide solution for 20 min followed by another washing step with deionized water (5 × 3 min). This was followed by another treatment with an aqueous 2% osmium tetroxide solution for additional 30 min at room temperature, in the dark and washing with deionized water (5 × 3 min). Block contrasting was conducted with a filtered, aqueous 1% uranyl acetate solution, overnight at 4°C, in the dark. On the next day, samples were washed with deionized water (5 × 3 min) and stained with lead aspartate (20 mmol lead nitrate in a 30 mmol L-aspartic acid solution, pH 5.5) for 30 min at 60°C. After thorough washing with deionized water (3 × 5 min), sections were dehydrated through an ascending series of ice-cold, aqueous ethanol dilutions (30%, 50%, 70%, 90%, 100%, each 5 min, 2 × 100%, anhydrous, each 10 min) before they were transferred into propylene oxide (2 × 10 min). Finally, the samples were infiltrated with an ascending series of Durcupan ACM (Sigma-Aldrich, Taufkirchen, Germany) in propylene oxide (2:1; 3:1, each for 1 hr and pure Durcupan ACM, overnight) before the sections were flat-embedded between two overhead projector foils, which in turn were placed between two microscopic glass slides and polymerized at 60°C for 2 days.

For the quantitative analysis of L4 SBs 3D-volume reconstructions were made based on z-stacks obtained using focused ion beam (FIB)

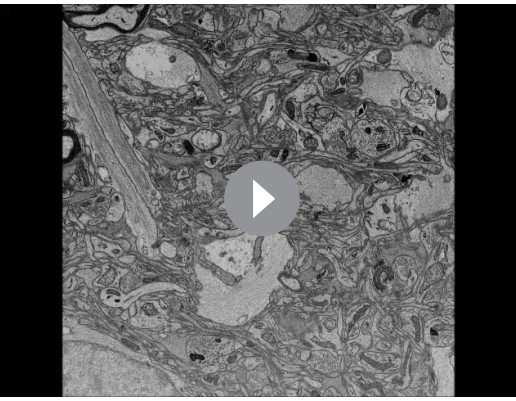

**Video 2.** FIB-SEM sequential video through L4 of the human TLN. Note the dynamic changes in the shape and size of dendritic and synaptic structures through the z-stack (250 single images).
https://elifesciences.org/articles/48373#video2

scanning electron microscopy (SEM). Based on the overall appearance of the sample, an area of interest was trimmed out of a flat-embedded section, using a 4 mm biopsy puncher, which was then glued onto a pre-polymerized resin block. Excess resin was removed around the tissue using a histology diamond knife on an ultramicrotome (UC7, Leica Microsystems, Vienna, Austria). The tissue sample was removed from the resin block with a razor blade and was then glued onto an SEM aluminum specimen stub using colloidal silver paste. The sample was dried in a vacuum chamber overnight, then sputter-coated with platinum/palladium for 15 s and finally placed into the FIB-SEM (Crossbeam 540, Carl Zeiss, Oberkochen, Germany) for 3D analysis.

A trench was milled with the FIB at 30 kV/30 nA, polishing of the surface was performed at 30kV/3nA and fine milling for data acquisition was performed at 30kV/7nA. The cross-section surface was imaged with an electron energy of 2keV and an electron beam current of 500 pA using an in-column energy-selective backscatter electron detector. The dwell time was 10 µs with line average 1. The pixel size in the XY-plane was 10 nm and the slice thickness (z-direction) was 50 nm yielding a voxel size of 10 nm x 10 nm x 50 nm. The image acquisition software Atlas 3D (Ver. 5.2.0.125, ZEISS, Oberkochen, Germany) allowed the automated collection of 3D SEM datasets using automated correction algorithms for drift, focus and astigmatism (*Video 2*). The advantage in using the FIB-SEM technique is three-fold: 1) a much higher throughput of different tissue samples at once; 2) definition of a much larger region of interest per sample and 3) increase of the z-dimensions of the individual samples. However, the disadvantage of this method is still the weaker resolution of single SVs compared to TEM. This approach, together with TEM, will be used in future studies for further image processing, 3D-volume reconstructions and subsequent data analysis.

## EM tomography of L4 SBs in the TLN

EM tomography was carried out on 200–300 nm thick sections cut from blocks prepared for ultrathin sectioning as described above. Sections were mounted on either pioloform-coated line or slot copper grids (Plano, Wetzlar, Germany) and were counterstained with uranyl acetate and lead citrate following a slightly modified staining protocol as described by *Reynolds (1963)*. Subsequently, sections were examined with a JEOL JEM 1400Plus, operating at 120 kV and equipped with a 4096 × 4096 pixels CMOS camera (TemCam-F416, TVIPS, Gauting, Germany). Tilt series were acquired automatically over an angular range of −60° to +60° at 1° degree increments using Serial EM (Ver. 3.58; *Mastronarde, 2005*). Stack alignment and reconstruction by filtered backprojection were carried out using the software package iMOD (Ver. 4.9.7; *Kremer et al., 1996*). Final reconstructions were ultimately filtered using a median filter with a window size of 3 pixels.

## CA of excitatory SBs in L4 of the TLN

CA was performed based on the structural parameters investigated (see *Table 2 - Source data 2*), to further identify different groups that are types of SBs by running a CA using MATLAB and Statistics Toolbox Release 2016b (The MathWorks, Inc, Natick, MA, USA; for details see *Yakoubi et al., 2019*). Then a zero-mean normalization was performed as the parameters had different units. This was followed by a PCA to reduce our large dataset to a smaller set of uncorrelated variables called PCs, but still containing most of the information in the original dataset. Subsequently, we performed a HCA on the simplified dataset composed of the PCs, as the original data were not labeled (*Figure 8 - Source data 2*; *Source code 1–3*, see also *Yakoubi et al., 2019*).

## Statistical analysis

The mean value ± SD, the median with the 1st and 3rd quartile, the $R^2$, the coefficient of variation (CV), skewness and variance were given for each parameter. The p-value was considered significant only if $p<0.05$. Boxplots were generated to investigate inter-individual differences for each structural parameter (*Figure 11 - Source data 2*). The non-parametric Kruskal-Wallis H-test analysis was computed, using InStat (GraphPad Software Inc, San Diego, CA, USA), as some of the analyzed parameters were not normally distributed as indicated by the skewness. Correlation graphs between several structural parameters were generated. To test the differences between L4 and L5 synaptic parameters, a Mann-Whitney u-test (unpaired; two-tailed) was performed using InStat (GraphPad Software Inc, San Diego, CA, USA).

## Acknowledgements

We thank our technicians Brigitte Marshallsay and Tayfun Palaz for their excellent technical assistance. Many thanks to PD Dr. Holger Jastrow for critically reading a prefinal version of the manuscript and helpful discussions and suggestions. The funding of Rachida Yakoubi by the DAAD and Joachim Lübke by the Helmholtz Society is very much acknowledged. The authors declare no financial and other conflict of interest.

## Additional information

### Funding

| Funder | Grant reference number | Author |
|---|---|---|
| Deutscher Akademischer Austauschdienst | | Rachida Yakoubi Joachim HR Lübke |
| Helmholtz-Gemeinschaft | Research Grant | Joachim HR Lübke |

The funders had no role in study design, data collection and interpretation, or the decision to submit the work for publication.

### Author contributions

Rachida Yakoubi, Kurt Sätzler, Data curation, Software, Formal analysis, Investigation, Methodology, Writing—original draft, Writing—review and editing; Astrid Rollenhagen, Conceptualization, Data curation, Software, Formal analysis, Investigation, Methodology, Writing—original draft, Writing—review and editing; Marec von Lehe, Investigation, Methodology; Dorothea Miller, Bernd Walkenfort, Conceptualization, Supervision, Validation, Investigation, Methodology, Writing—original draft, Writing—review and editing; Mike Hasenberg, Investigation, Methodology, Writing—original draft, Writing—review and editing; Joachim HR Lübke, Conceptualization, Data curation, Software, Formal analysis, Supervision, Validation, Investigation, Methodology, Writing—original draft, Writing—review and editing

### Author ORCIDs

Joachim HR Lübke (iD) https://orcid.org/0000-0002-4086-3199

### Ethics

Human subjects: The consent of the patients was obtained and all experimental procedures were approved by the Ethical Committees of the Rheinische Friedrich-Wilhelms-University/University Hospital Bonn (ethic votum of the Medical Faculty to Prof. Dr. med. Johannes Schramm and Prof. Dr. rer. nat. Joachim Lübke, Nr. 146/11), and the University of Bochum (ethic votum of the Medical Faculty to PD Dr. med. Marec von Lehe and Prof. Dr. rer. nat. Joachim Lübke, Reg. No. 5190-14-15; ethic votum of the Medical Faculty to Dr. med. Dorothea Miller and Prof. Dr. rer. nat. Joachim Lübke, Reg. No. 17-6199-BR), and the EU directive (2015/565/EC and 2015/566/EC) concerning working with human tissue.

### Decision letter and Author response

Decision letter https://doi.org/10.7554/eLife.48373.sa1
Author response https://doi.org/10.7554/eLife.48373.sa2

## Additional files

### Supplementary files

- Source code 1. Matlab code for CA of all synaptic parameters investigated.
- Source code 2. Matlab code for CA of PreAZs and PSDs.
- Source code 3. Matlab code for CA of pools of SVs.

- Source data 1. Original data for synaptic density measurements.
- Source data 2. Original data of all synaptic parameters analyzed.
- Transparent reporting form

### Data availability

All datasets have been provided as source files in the article.

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
