## [Decision Letter]

**Acceptance summary:**

Chemical synapses as the main sites of neuronal interaction in mammals have been studied extensively. The detailed structural setup of excitatory synapses in the human neocortex, however, is still much less explored. Here, the authors report a detailed electron-microscopic analysis of the structural features of synapses in layer 4 of human temporal lobe neocortexThe tissue was obtained from biopsies during epilepsy surgery aimed at deeper brain structures. The authors report remarkably large synaptic vesicle pools, that are even larger than those observed in the synaptic boutons of human synapses from layer 5, which are at the same time about 5-fold larger in volume. This data emphasizes the need to analyse human cortical tissue for understanding the commonalities and differences to data obtained from rodents.

**Decision letter after peer review:**

Thank you for submitting your article "Ultrastructural heterogeneity of human layer 4 excitatory synaptic boutons in the adult temporal lobe neocortex" for consideration by *eLife*. Your article has been reviewed by three peer reviewers, and the evaluation has been overseen by a Reviewing Editor and Eve Marder as the Senior Editor. The following individuals involved in review of your submission have agreed to reveal their identity: Kathleen Rockland (Reviewer #1).

The reviewers have extensively discussed the reviews with one another and the Reviewing and Senior Editors, and the Reviewing Editor has drafted this decision to help you prepare a revised submission.

The manuscript reports an ultrastructural description of synapse composition in layer 4 of human cortex obtained from epilepsy surgery samples. The data are of very high quality and provide a reference for layer 4 synaptic composition in human.

While the relevance of data from the human brain was seen by all reviewers, the reviewers were concerned about the relationship of the presented data to a previous publication by the same group about synapses in L5 of human neocortex (Yakoubi et al., 2018).

The reviewers therefore agreed to request an extensive revision of the manuscript that should ensure:

- a clear and insightful comparison of the presented data to the L5 data, and to the literature (see detailed points in the reviews below). This may include the need for additional analyses, see below.

- a more elaborate explanation of the significance of data from L4, especially in the temporal lobe.

- a much extended methodological description that provides all details required for comparing the results to previous work. Again, the reviews below describe the missing detail.

*Reviewer #1:*

Anatomical data in human brain are still frustratingly limited, and in this regard, the current manuscript can enjoy an assumption of relevance. Added to that, Dr. Lubke and colleagues have an outstanding record of technical expertise. Thus, I am very positive as to relevance and quality of data. My comments address mainly points of style and clarity.

1) There is a very relevant previous paper from this group: Yakoubi et al., 2018. My suggestion is that this should be more prominently referenced, certainly in the Introduction and Discussion, and possibly the Abstract.

2) After my first reading, I felt the Discussion was perhaps too long and "fluffy." But I have no suggestions for shortening, and am overall inclined to think that the main problem is one of emphasis and wording at various points. Thus:

– Subsection “Relevance and implications of the density of synaptic contacts measurements” paragraph two, disparity with Alonso-Nanclares et al. This is important, and I think needs some more careful re-wording.

–Paragraph three of the same section gender difference. This is important, but the interpretation is ineffective. The Authors might state that underlying reasons are difficult to determine, but they can speculate xyz. One might also think of hormonal/hypothalamic differences? and/or environmental? and/or "emotional" as via the amygdala, but further data are obviously needed.

– Subsection “Shape and size of AZs at excitatory L4 SBs in the human TLN”; There are relevant, not cited data from NHP: Amaral on amygdala-cortical synapses; Kevan Martin on synapses in MT. The Authors might consider some slight expansion of this section.

– Subsection “Other important structural components of synaptic complexes in L4 of the human TLN”. I do not think it is successful to lump the three observations on spine apparatus, mitochondria, and astrocytes. I urge the Authors to consider separate headings (or subheadings).

– Subsection “Functional relevance”: Temporal cortex is higher order, not primary, or early sensory.Thus, layer 4 is a convergent input layer for both thalamocortical and corticocortical inputs. As stated in this version, it is confusing.

– Similarly, "first station" does not strictly apply, since we are out of the primary sensory areas.

3) Figures. These are overall very nice. However, the laminar boundaries seem a bit inexact, mainly because the indicating lines are not parallel to the pia. For Figure 2, layer 2 (about the same thickness as layer 1?) should again be distinguished from layer 3.

*Reviewer #2:*

The study by Yakoubi et al. examines the ultrastructure of human brain tissue from the region of layer 4 the temporal cortex. The analysis has included a number of samples from the different patients undergoing surgery, both male and female. The paper appears to be a follow-on study published in 2018 in Cerebral Cortex in which the same authors studied layer 5 of human temporal cortex. The Results section gives a detailed set of figures showing a number of parameters concerning the morphology of synaptic boutons.

The quality of the electron microscopy and the analyses are clear and precise, however, I have a few concerns about some of the interpretations, and the overall significance of the results. I appreciate this piece of work as a careful analysis of bouton structure in the human brain. They are interesting, however, how interesting to a broad audience is questionable. One reason is that it's not obvious how comparisons of certain parameters such as synapse sizes and vesicle pool sizes with other regions in other species can say much about functional differences, or any about the rules of connectivity of plasticity. Comparisons within the same tissue would perhaps be more valuable, but this is just a single snapshot, and interpretations from this structural data in terms of the plasticity of connectivity is perhaps stretching it somewhat.

I have some questions about the images shown and the analysis. Figure 1 shows some electron micrographs of gap junctions. However, It's not clear as to whether these are gap junctions or not. In part F (Figure 1) it would appear that there is a segment of endoplasmic reticulum against the membrane and I wonder how well the authors are able to identify a cell junction. It's not evident in the image shown. Also, the characteristic periodicity in staining along the length of a gap junction is missing, in either of the images.

In Figure 2 some examples of axonal boutons are shown. However, are these just the glutamatergic type, or are any others shown? There is little mention of which boutons are analyzed, and it's also not clear how the authors define where a bouton starts and the axon ends. What criteria did they use to define just the bouton? In the example shown in Figure 2D this single bouton extends over several microns and appears to be multisynaptic, and very different in morphology from the others shown, at least in the 2D images. How were the synapses on multi-synaptic boutons considered.

In Table 2 the authors have measured the size of the synaptic cleft, but it's not clear why. Is there a functional significance to this analysis, and the results?

In several places, there are images showing possible vesicles emerging from the mitochondria and referred to as 'mitochondrial-derived vesicles'. The evidence for these type of structures at the EM level is weak, and these anecdotal images add little to the paper. To include this would require at least some verification that this is not some artifact of fixation.

In Figure 9, the authors indicate the presence of an astrocytic element close to a synapse. What criteria have been used to identify this element as astrocytic? Could this not be a microglial process?

The revisions should include more details of the methods. The authors give little information as to how extensively the synaptic densities were measured in each sample. How many synapses were counted overall? I also found it odd that they make no compensation for tissue shrinkage, stating instead that the paper by Korogod showed little change, when in fact it showed the opposite. This makes it more difficult to compare with other studies. On this point, they cite the paper of Mayhew, 1996 that proposes expressing synapse number as a function of synapses per neuron, or per brain region rather than per unit volume, which they clearly do not apply here. It would, of course, make it difficult to compare with their previous work, but a better explanation should at least be given.

*Reviewer #3:*

This study examines ultrastructural features of spines in layer 4 neurons of the temporal lobe of human cortex. The work appears to be generally well done. However, the study may be viewed as incremental because of similar, previously published EM analysis of spines in human cortex, from the same group:

Quantitative Three-Dimensional Reconstructions of Excitatory Synaptic Boutons in Layer 5 of the Adult Human Temporal Lobe Neocortex: A Fine-Scale Electron Microscopic Analysis. Yakoubi R, Rollenhagen A, von Lehe M, Shao Y, Sätzler K, Lübke JHR. Cerebral Cortex, https://doi.org/10.1093/cercor/bhy146. Published: 21 June 2018.

In particular, the 2018 Cerebral Cortex publication examined human layer 5 neurons whereas the present study examines human layer 4 neurons. Similarities in the numbers of synaptic vesicles reported in the 2018 paper and the present study (1800 vs 1500) and the readily-releasable pool (20 vesicles in spines in each of the studies) and active zone sizes are notable. So, the present study is highly related to published work by this group, focusing on different neurons of the human temporal lobe, a short distance away, and many of the conclusions are the same, including that synapses and their vesicle compositions are much larger in human cortex than other species. For this reviewer, the advance in the present study is incremental, and the work would be better suited for a specialized journal.

[Editors' note: further revisions were requested prior to acceptance, as described below.]

Thank you for resubmitting your work entitled "Ultrastructural heterogeneity of human layer 4 excitatory synaptic boutons in the adult temporal lobe neocortex" for further consideration at *eLife*. Your revised article has been favorably evaluated by Eve Marder (Senior Editor), a Reviewing Editor, and two reviewers.

The manuscript has been improved but there are some remaining issues that need to be addressed before acceptance, as outlined below:

The reviewers and editor were disappointed that the authors did not make more efforts to address the reviewers' concerns, in particular since some of them were merely requests for additional context and discussion, i.e. textual additions. Below reviews give a very constructive description of what would be required to make this manuscript a worthwhile addition to the previously published L5 data.

Furthermore, the issue of tissue shrinkage needs to be rightfully addressed, with proper references to the literature, as outlined by reviewer 2.

Finally, the request for clarification of the definition of gap junctions should be plausible within the context of this paper (see comments of reviewer 2 on Figure 1F,G).

We will be happy to evaluate a final revision of this manuscript, but do request that the points raised are faithfully and thoroughly addressed in the final revision.

*Reviewer #1:*

I continue to admire, in this revised manuscript, the quality of the research and the potential significance of the results. At the same time, I have rarely been so disappointed with the revisions, which in this case are truly minimalist. My particular requests had been for a careful comparison with previous work from this laboratory on layer 5 (Yaboubi et al., 2019). "We previously reported…. In ongoing work along these lines, we next… Our motivation…." There are only a slight additional reference to this 2019 paper in the revised manuscript, with the consequence that the present extension into layer 4 is left to dangle as little more than a descriptive add-on. I acknowledge that further discussion of layer specific microcircuitry implications risks to be "overly speculative," but there are several tactics the Authors could have taken; for example, to refer to and discuss other reports on human specializations (i.e., Mansvelder), or differences in monkey and rodent (Luebke and colleagues). The stage must be carefully set to engage the reader, and without this, the very fine results, to repeat, come across as merely "descriptive." A "meaty" paragraph or more, in the Introduction and Discussion are essential – not just a casual reference.

*Reviewer #2:*

This re-submission is well presented with very high-quality EM imaging. The previous comments have been addressed and I appreciate the authors' qualification of most of the points raised.

I only have two outstanding points, also brought up before. I fully recognize the senior author's expertise in electron microscopy and thank him for including some previous images of gap junctions. They are beautiful. However, I'm still unclear about what is shown in Figure 1 but accept I could be missing something, or there is a lack of pixels in this copy. In part F, I see 4 parallel membranes and wonder where the junctional complex is, and how this is distinguished. In part G, I see three parallel membranes and it’s not clear which part represents the gap between the two apposed membranes.

The final point is concerning the tissue shrinkage. I fully accept the difficulties in making comparisons due to the wide differences between fixation protocols. That would be the advantage of normalizing the value per neuronal cell body or axonal length. However, the authors continue to state that, 'overall density of synaptic contacts was not significantly different…' when comparing aldehyde or cryo-fixed and substituted tissues. The paper they refer to, Korogod et al., 2015, shows a significant difference in synapse density between the two fixation methods.

---

## [Author Response]

Reviewer #1:Anatomical data in human brain are still frustratingly limited, and in this regard, the current manuscript can enjoy an assumption of relevance. Added to that, Dr. Lubke and colleagues have an outstanding record of technical expertise. Thus, I am very positive as to relevance and quality of data. My comments address mainly points of style and clarity.1) There is a very relevant previous paper from this group: Yakoubi et al., 2018. My suggestion is that this should be more prominently referenced, certainly in the Introduction and Discussion, and possibly the Abstract.

We agree with the reviewer to quote Yakoubi et al., 2019 (now published) more often in the Introduction, Results and Discussion where appropriate, but not in the Abstract due to word limitation.

2) After my first reading, I felt the Discussion was perhaps too long and "fluffy." But I have no suggestions for shortening, and am overall inclined to think that the main problem is one of emphasis and wording at various points. Thus:– Subsection “Relevance and implications of the density of synaptic contacts measurements” paragraph two, disparity with Alonso-Nanclares et al. This is important, and I think needs some more careful re-wording.

We have rephrased the text as suggested by the reviewer.

–Paragraph three of the same section gender difference. This is important, but the interpretation is ineffective. The Authors might state that underlying reasons are difficult to determine, but they can speculate xyz. One might also think of hormonal/hypothalamic differences? and/or environmental? and/or "emotional" as via the amygdala, but further data are obviously needed.

We have added a short text and a reference to further explain reasons underlying gender-specific differences in density of synaptic contacts as suggested. However, we would like to point out that this is not the main focus of the study, but a very interesting finding to be included.

– Subsection “Shape and size of AZs at excitatory L4 SBs in the human TLN”; There are relevant, not cited data from NHP: Amaral on amygdala-cortical synapses; Kevan Martin on synapses in MT. The Authors might consider some slight expansion of this section.

We have added the missing references where appropriate and extended the section as requested.

– Subsection “Other important structural components of synaptic complexes in L4 of the human TLN”. I do not think it is successful to lump the three observations on spine apparatus, mitochondria, and astrocytes. I urge the Authors to consider separate headings (or subheadings).

We have introduced the subheadings as suggested.

– Subsection “Functional relevance”: Temporal cortex is higher order, not primary, or early sensory. Thus, layer 4 is a convergent input layer for both thalamocortical and corticocortical inputs. As stated in this version, it is confusing.

We agree with the reviewer on this point. We have changed the text accordingly where appropriate (see Introduction and Discussion) and added a review for clarity.

– Similarly, "first station" does not strictly apply, since we are out of the primary sensory areas.

We have rephrased the text upon the role of L4 in the TLN throughout the text.

3) Figures. These are overall very nice. However, the laminar boundaries seem a bit inexact, mainly because the indicating lines are not parallel to the pia. For Figure 2, layer 2 (about the same thickness as layer 1?) should again be distinguished from layer 3.

We have changed Figure 1 and Figure 1-figure supplement 1 to better indicate the layer thickness and boundaries as suggested.

Reviewer #2:The study by Yakoubi et al. examines the ultrastructure of human brain tissue from the region of layer 4 the temporal cortex. The analysis has included a number of samples from the different patients undergoing surgery, both male and female. The paper appears to be a follow-on study published in 2018 in Cerebral Cortex in which the same authors studied layer 5 of human temporal cortex. The Results section gives a detailed set of figures showing a number of parameters concerning the morphology of synaptic boutons.The quality of the electron microscopy and the analyses are clear and precise, however, I have a few concerns about some of the interpretations, and the overall significance of the results. I appreciate this piece of work as a careful analysis of bouton structure in the human brain. They are interesting, however, how interesting to a broad audience is questionable. One reason is that it's not obvious how comparisons of certain parameters such as synapse sizes and vesicle pool sizes with other regions in other species can say much about functional differences, or any about the rules of connectivity of plasticity. Comparisons within the same tissue would perhaps be more valuable, but this is just a single snapshot, and interpretations from this structural data in terms of the plasticity of connectivity is perhaps stretching it somewhat.

We disagree with the reviewer upon the overall significance of the results and would like to quote reviewer 1: “Anatomical data in human brain are still frustratingly limited, and in this regard, the current manuscript can enjoy an assumption of relevance”. First, it is still very controversially discussed whether experimental data obtained in animals can be transferred 1:1 to humans. Since there are not many quantitative data available in humans, the comparison with experimental data obtained from various animal species, also limited and not always coherent, is currently a good option to assess similarities and differences in synaptic structures. Our final aim is to describe the layer-specific synaptic organization of the human neocortex, exemplified for the temporal lobe neocortex. Such data are not at all available and thus represent an important add-on to what is already known about the quantitative geometry of CNS synapses. In the reviewer’s comments of already published work from our group it is always stated that such data are long expected and required to better understand the role of these structures in synaptic transmission and plasticity.

The reviewer is right that a comparison with the same tissue is valuable. Hence we have included the already published data from layer 5 excitatory synaptic boutons to Table 2 and discussed the findings in the human temporal lobe in more detail (see text).

We again disagree with the reviewer on his last point. For example differences in the density of synaptic contacts in a certain volume of a given brain region can help to predict connectivity rates in the corresponding region. Moreover, it has been demonstrated by meanwhile several studies that structural data of synaptic boutons, such as the number, size and shape of the active zones and the organization and size of the three pools of synaptic vesicles, involvement of astrocytes and mitochondria may allow prediction about the release machinery, mode and probability of neurotransmitter release and thus about synaptic transmission and plasticity. For example, it has been demonstrated that the size of the active zones is positively correlated to the mode and probability of release (for example Matz et al., 2010; Freche et al., 2011; Holderith et al., 2012). The quantitative structural analysis of the three functionally defined vesicle pools also allow prediction about synaptic strength, efficacy and modulation of synaptic plasticity (for example Denker and Rizzoli, 2010; Imig et al., 2014; Schikorski, 2014; reviewed by Rizzoli and Betz, 2004, 2005). All the other structural parameters mentioned above were thoroughly discussed in the manuscript.

Meanwhile such data as analyzed here provide the basis for numerical or Monte Carlo simulations of various synaptic parameters, for example neurotransmitter diffusion via the synaptic cleft, that at least in humans are still inaccessible to experiment. To further demonstrate the value of such data, we were asked by several computational neuroscientists to provide our data to perform realistic simulations.

I have some questions about the images shown and the analysis. Figure 1 shows some electron micrographs of gap junctions. However, It's not clear as to whether these are gap junctions or not. In part F (Figure 1) it would appear that there is a segment of endoplasmic reticulum against the membrane and I wonder how well the authors are able to identify a cell junction. It's not evident in the image shown. Also, the characteristic periodicity in staining along the length of a gap junction is missing, in either of the images.

We disagree with the reviewer´s opinion. We would like to point out that the senior author of the paper has a more than 30 years of expertise in electron microscopy, and is thus able to identify gap junctions and other membrane specializations. In Figure 1F we have followed the gap junction over consecutive sections to make sure that is indeed a gap junction and took the best image for publication. In Figure 1G a higher magnification of a gap junction was taken to better visualize the typical three-layered structure of this membrane specialization that makes it also distinguishable from desmosomes and tight junctions. Further examples of gap junctions, meeting the criteria as used by us, are found in numerous text books and publications (for example see Figure 6 from Vervaeke et al., 2010).

In Figure 2 some examples of axonal boutons are shown. However, are these just the glutamatergic type, or are any others shown? There is little mention of which boutons are analyzed, and it's also not clear how the authors define where a bouton starts and the axon ends. What criteria did they use to define just the bouton? In the example shown in Figure 2D this single bouton extends over several microns and appears to be multisynaptic, and very different in morphology from the others shown, at least in the 2D images. How were the synapses on multi-synaptic boutons considered.

We disagree with the reviewer because it is mentioned in the title and throughout the text that we have focused our analysis on excitatory synaptic boutons in layer 4 TLN. For clarity we have added a sentence to the Results section and to the figure legends. We agree with the reviewer about the definition of where the bouton starts and axon ends and added a text in material and methods. Regarding Figure 2D, does the reviewer mean the multisynaptic boutons in Figure 2B and C? If so, we would like to mention that the majority (~97%) of excitatory synaptic boutons in L4 contained a single release site equivalent to the synapse, the term used by the reviewer. The remaining contained two to three synapses as exemplified in Figure 2 B and C which were not differently handled in our analysis. For clarity we have added a text in Materials and methods under the chapter: 3D-volume reconstructions and quantitative analysis of excitatory L4 SBs.

In Table 2 the authors have measured the size of the synaptic cleft, but it's not clear why. Is there a functional significance to this analysis, and the results?

The width of the synaptic cleft is of importance for the temporal and spatial transient increase of the glutamate concentration in the cleft, reversible binding of glutamate to appropriate glutamate receptors and the vertical but also horizontal diffusion and eventual uptake of glutamate out of the cleft by fine astrocytic processes. To a large extent, these processes are governed by the geometry and width of the synaptic cleft. However, how and to what extent a larger cleft width contribute to differences in synaptic transmission has to be directly demonstrated by electrophysiology or simulations (Freche et al., 2011). The synaptic cleft measurements will be needed for future simulation studies in our lab together with the Jülich Supercomputing Centre. In our past studies, this measurement was requested by other reviewers because of its relevance to synaptic transmission and plasticity. We have also added a small paragraph for further explanation in Materials and methods under chapter: 3D-volume reconstructions and quantitative analysis of excitatory L4 SBs.

In several places, there are images showing possible vesicles emerging from the mitochondria and referred to as 'mitochondrial-derived vesicles'. The evidence for these type of structures at the EM level is weak, and these anecdotal images add little to the paper. To include this would require at least some verification that this is not some artifact of fixation.

Meanwhile numerous publications have substantiated the existence of mitochondrial-derived vesicles using different methods and electron microscopic protocols (for example: Sugiura et al., 2014 EMBO Journal; reviewed by Andrade-Navarro et al., 2009). Hence these structures cannot be regarded as an artefact of fixation. Even anecdotal and unfrequently observed in our samples, these structures are an interesting finding that add information to describe synaptic boutons in humans.

In Figure 9, the authors indicate the presence of an astrocytic element close to a synapse. What criteria have been used to identify this element as astrocytic? Could this not be a microglial process?

We have identified astrocytic processes according to established criteria introduced in the famous textbook about the ultrastructure of the nervous system: neurons and supporting cells by Peters et al., 1991, and later adopted and published by Ventura and Harris, 1999.

Astrocytic processes were identified by their irregular, stellate shape and by the presence of glycogen granules and bundles of intermediate filaments in a relatively clear cytoplasm (see also Peters et al., 1991). In addition to astrocytes, we have also observed microglial cells and macrophages, especially near to degenerating neurons. Microglial cells are always distinguishable from astrocytes and their fine processes, by their different shape and appearance at the electron microscopic level. In addition, their occurrence was infrequent and varied substantially between tissue samples. Hence, we have never seen fine processes of microglial cells near synaptic complexes or establishing direct contact with a bouton. In contrast there is structural evidence of the existence of such contacts between astrocytes and their target structures as shown for example for the so-called tripartite synapse in the hippocampus and supraoptic nucleus of the hypothalamus. We have added a short paragraph to the Results sections, under the chapter: Astrocytic coverage of L4 SBs in the human TLN.

The revisions should include more details of the methods. The authors give little information as to how extensively the synaptic densities were measured in each sample. How many synapses were counted overall? I also found it odd that they make no compensation for tissue shrinkage, stating instead that the paper by Korogod showed little change, when in fact it showed the opposite. This makes it more difficult to compare with other studies. On this point, they cite the paper of Mayhew, 1996, that proposes expressing synapse number as a function of synapses per neuron, or per brain region rather than per unit volume, which they clearly do not apply here. It would, of course, make it difficult to compare with their previous work, but a better explanation should at least be given.

We disagree with the reviewer upon our density measurements and not compensating for tissue shrinkage. For the density measurements we followed established and standardized protocols used by other scientists that performed such an analysis as cited in Materials and methods under chapter: Stereological estimation of the density of synaptic contacts. In this context the reviewer misinterpreted, in our opinion, that “Mayhew 1996” proposes expressing synapse number as a function of synapses per neuron, or per brain region rather than per unit volume, because: the physical dissector method implies already calculation per unit volume as also indicated by the equation used here (see Materials and methods). This led to a synaptic density per unit volume which could be then compared to other published data evaluated the same way.

Regarding tissue shrinkage, there is still a controversial and ongoing discussion whether structural experimental data should be corrected for shrinkage. We would like to point out here that different subelements in the neuropil, such as somata, dendrites, axons and synaptic vesicles, show different patterns of shrinkage. Hence, a general shrinkage factor is hard to apply. Thus, in numerous quantitative studies, shrinkage is not corrected. If a correction for shrinkage was applied, various different methods for estimating the tissue shrinkage factor were used, and thus, the experimental data corrected for shrinkage are not comparable. Furthermore, tissue shrinkage could not only be related to the fixation process, but could be also influenced by various other factors like differences in the protocols used for electron microscopy, for example the use of cacodylate vs phosphate buffer; ethanol vs. acetone dehydration and embedding media.

After again thoroughly reading the Korogod-paper, the following assumptions can be made: the overall brain volume is reduced in chemical fixation due to the reduction of the extracellular space and glial volume. However, no significant difference was found in neuronal elements, such as dendrites and axons. Moreover, no significant difference was found in the overall synaptic density, only for symmetric synapses which are not relevant for the present study as we analyzed asymmetric (excitatory) synapses.

Surprisingly, the number of synaptic vesicles in the readily releasable pool was somewhat lower in cryo-substituted material when compared to chemical fixation (but see Sätzler et al., 2002; Zhao et al., 2012b), which is the only difference to our findings.

Finally, we would like to quote the final sentence of the Korogod-paper: “Large-scale preservation for ultrastructural analysis will therefore continue to rely on chemical fixation approaches”.

However, we have extended the paragraph regarding tissue shrinkage in Materials and methods section under the chapter: 3D-volume reconstructions and quantitative analysis of excitatory L4 SBs.

Reviewer #3:This study examines ultrastructural features of spines in layer 4 neurons of the temporal lobe of human cortex. The work appears to be generally well done. However, the study may be viewed as incremental because of similar, previously published EM analysis of spines in human cortex, from the same group:Quantitative Three-Dimensional Reconstructions of Excitatory Synaptic Boutons in Layer 5 of the Adult Human Temporal Lobe Neocortex: A Fine-Scale Electron Microscopic Analysis. Yakoubi R, Rollenhagen A, von Lehe M, Shao Y, Sätzler K, Lübke JHR. Cerebral Cortex, https://doi.org/10.1093/cercor/bhy146. Published: 21 June 2018.In particular, the 2018 Cerebral Cortex publication examined human layer 5 neurons whereas the present study examines human layer 4 neurons. Similarities in the numbers of synaptic vesicles reported in the 2018 paper and the present study (1800 vs 1500) and the readily-releasable pool (20 vesicles in spines in each of the studies) and active zone sizes are notable. So, the present study is highly related to published work by this group, focusing on different neurons of the human temporal lobe, a short distance away, and many of the conclusions are the same, including that synapses and their vesicle compositions are much larger in human cortex than other species. For this reviewer, the advance in the present study is incremental, and the work would be better suited for a specialized journal.

We disagree with this reviewer’s opinion about the non-suitability of this paper for *eLife*, because of the fact that it only considered as an extension to a previously published work and the conclusions are the same. Again we would like to quote reviewer 1 “Anatomical data in human brain are still frustratingly limited, and in this regard, the current manuscript can enjoy an assumption of relevance”.

Our submission is further substantiated by the need of such structural data for functional studies (for example see Seeman et al., 2018 published in *eLife*) and computational neuroscientists to perform different simulations based on realistic values of several structural synaptic parameters. As already stated in our reply to reviewer 2 we were asked to provide these data for further modelling. The final aim as already raised above is to describe the synaptic organization of the neocortex exemplified for the temporal lobe neocortex. To do so, it is important to describe, at the subcellular level, each cortical layer individually to look for similarities and differences between layers. The synaptic organization in each layer may contribute to explain the computational properties of neurons in intra- and translaminar networks, for example constituting the network of a cortical column.

Finally, synapses are key elements in any given network of the brain. Detailed information about their structural composition is required for an improved understanding of the function of the brain in health and disease.

[Editors' note: further revisions were requested prior to acceptance, as described below.]

The reviewers and editor were disappointed that the authors did not make more efforts to address the reviewers' concerns, in particular since some of them were merely requests for additional context and discussion, i.e. textual additions. Below reviews give a very constructive description of what would be required to make this manuscript a worthwhile addition to the previously published L5 data.Furthermore, the issue of tissue shrinkage needs to be rightfully addressed, with proper references to the literature, as outlined by reviewer 2.Finally, the request for clarification of the definition of gap junctions should be plausible within the context of this paper (see comments of reviewer 2 on Figure 1F,G).We will be happy to evaluate a final revision of this manuscript, but do request that the points raised are faithfully and thoroughly addressed in the final revision.Reviewer #1:I continue to admire, in this revised manuscript, the quality of the research and the potential significance of the results. At the same time, I have rarely been so disappointed with the revisions, which in this case are truly minimalist. My particular requests had been for a careful comparison with previous work from this laboratory on layer 5 (Yaboubi et al., 2019). "We previously reported…. In ongoing work along these lines, we next… Our motivation…." There are only a slight additional reference to this 2019 paper in the revised manuscript, with the consequence that the present extension into layer 4 is left to dangle as little more than a descriptive add-on. I acknowledge that further discussion of layer specific microcircuitry implications risks to be "overly speculative," but there are several tactics the Authors could have taken; for example, to refer to and discuss other reports on human specializations (i.e., Mansvelder), or differences in monkey and rodent (Luebke and colleagues). The stage must be carefully set to engage the reader, and without this, the very fine results, to repeat, come across as merely "descriptive." A "meaty" paragraph or more, in the Introduction and Discussion are essential – not just a casual reference.

We again thank Dr. Kathleen Rockland for her helpful suggestions to improve the manuscript. We have now hopefully addressed her request for a ‘meaty’ paragraph in the Introduction and Discussion where we directly compared human L4 and L5 in the TLN and point to our final goal to describe the synaptic organization, layer by layer of the cortical column (Introduction). We also gave a perspective at the end of the discussion that we plan to also work on epileptic neocortical tissue. However, we would like to point out that we improved our manuscript by directly comparing our results of L4 and L5 in humans and other experimental animal including NHPs; these comparisons are now highlighted in blue throughout the text, including the functional relevance chapter. However, we already discussed structural and functional layer-specific differences in layer 4 synaptic connections of the rodent (Feldmeyer et al., 1999, 2002, Silver et al., 2003) and between rodents and humans (Seemann et al. 2018). The same holds true for our results about bouton geometry and AZ size with findings in rodents and non-human primates (see papers by Hsu et al., 2017 and Bopp et al., 2017). Since the reviewer already pointed out that the discussion is already long we have now just added a short hopefully ‘meaty’ chapter about future perspectives to work on the human TLN by directly comparing results in ‘health’ and disease.

Reviewer #2:This re-submission is well presented with very high-quality EM imaging. The previous comments have been addressed and I appreciate the authors' qualification of most of the points raised.I only have two outstanding points, also brought up before. I fully recognize the senior author's expertise in electron microscopy and thank him for including some previous images of gap junctions. They are beautiful. However, I'm still unclear about what is shown in Figure 1 but accept I could be missing something, or there is a lack of pixels in this copy. In part F, I see 4 parallel membranes and wonder where the junctional complex is, and how this is distinguished. In part G, I see three parallel membranes and it’s not clear which part represents the gap between the two apposed membranes.

First, we would like to point that we followed the definition of a Gap-junction by well-established criteria set by others. Per definition, Gap-junctions are specialized direct intercellular connections between two neurons, which allows various molecules, ions and electrical impulses to directly pass through a regulated gate between the cells. One Gap-junction channel is composed of two connexons (or hemichannels), which connect across the intercellular space. At the ultrastructural level Gap-junction plaques were confirmed to have variable composition being home to connexon and non-connexin, but display a typical-three layered structure with two distinct and apparent band of connexons separated by two lipid layers spaced by another but thinner band of proteins. However, Gap-junctions are variable in size and the third thinner band can be disrupted even Gap-junctions can be periodically interrupted.

We have used this criteria to identify these membrane specializations in our serial sections as exemplified in Figure 4F and G. Furthermore, we were able to follow Gap-junctions over several consecutive ultrathin sections where these structures markedly change in shape and size and were disrupted. Moreover, it is rare to see a long-range continuous band connecting the two neurons. In addition, we also asked for the expertise of two colleagues, Prof. Karl Zilles (FZJ Jülich) and Prof. Holger Jastrow (Medical Research Center, University of Duisburg-Essen), which are experts in the field of subcellular structures at the EM level. They both confirmed our opinion that what we show in Figure 4F, G are indeed Gap-junctions.

Furthermore, we would like to mention that we have a paper published in Frontiers in Neuroanatomy where we described Gap-junctional coupling between clusters of neurons in the reeler mouse neocortex, and identified Gap-junctions by the same criteria where our results were not questioned by the reviewers.

Finally we would like to stress that Gap-junctional coupling was an interesting side finding but not the main focus of our study. Thus we hope that the reviewer is now convinced with our reply.

The final point is concerning the tissue shrinkage. I fully accept the difficulties in making comparisons due to the wide differences between fixation protocols. That would be the advantage of normalizing the value per neuronal cell body or axonal length. However, the authors continue to state that, 'overall density of synaptic contacts was not significantly different…' when comparing aldehyde or cryo-fixed and substituted tissues. The paper they refer to, Korogod et al., 2015, shows a significant difference in synapse density between the two fixation methods.

We thank the reviewer for his comment that defining tissue shrinkage is difficult to achieve and to compare due to the wide differences between fixation and embedding protocols. We would like to mention again that there is still an intensive ongoing, but controversial discussion whether experimental data should be corrected for shrinkage. We and many other scientists that face this discussion decided not to correct our experimental data because of numerous uncertainties during fixation and embedding that could lead to a false correction in shrinkage. As already stated in our first rebuttal we believe that certain structures shrink differently during the fixation and embedding process.

Concerning the density of synaptic contacts the reviewer is correct in his statement that significant differences in synaptic density exist between the two fixation methods used in the Korogod-paper. Comparing the two given values the difference was ~14%, however, with a significance value of just under 0.05 (0.042). Nevertheless, we have extended the chapter about tissue shrinkage in Materials and Methods to point out the difficulties one has with exactly determining tissue shrinkage.